# Seasonal distribution and drivers of surface fine particulate matter and organic aerosol over the Indo-Gangetic Plain

Caterina Mogno[1], Paul I. Palmer[1,2], Christoph Knote[3], Fei Yao[1], and Timothy J. Wallington[4]

[1]School of GeoSciences, University of Edinburgh, Edinburgh, UK
[2]National Centre for Earth Observation, University of Edinburgh, Edinburgh, UK
[3]Model-Based Environmental Exposure Science (MBEES), Faculty of Medicine, University of Augsburg, Germany
[4]Research & Advanced Engineering, Ford Motor Company, Dearborn, MI, 48121-2053, USA

**Correspondence:** Caterina Mogno (c.mogno@ed.ac.uk)

**Abstract.**

**The Indo-Gangetic Plain (IGP) is home to 9% of the global population** and is responsible for a large fraction of agricultural crop production in Pakistan, India, and Bangladesh. Levels of fine particulate matter (mean diameter <2.5 microns, $PM_{2.5}$) across the IGP often exceed human health recommendations, making cities across the IGP among the most polluted in the world. Seasonal changes in the physical environment over the IGP are dominated by the large-scale South Asian monsoon system that dictates the timing of agricultural planting and harvesting. We use the WRF-Chem model to study the seasonal anthropogenic, pyrogenic, and biogenic influences on fine particulate matter and its constituent organic aerosol (OA) over the IGP that straddles Pakistan, India, and Bangladesh during 2017/2018. We find that surface air quality during pre-monsoon (March—May) and monsoon (June—September) seasons is better than during post-monsoon (October—December) and winter (January—February) seasons, but all seasonal mean values of $PM_{2.5}$ still exceed the recommended levels, so that air pollution is a year-round problem. Anthropogenic emissions influence the magnitude and distribution of $PM_{2.5}$ and OA throughout the year, especially over urban sites, while pyrogenic emissions result in localized contributions over the central and upper parts of IGP in all non-monsoonal seasons, with the highest impact during post-monsoon seasons that correspond to the post-harvest season in the agricultural calendar. Biogenic emissions play an important role in the magnitude and distribution of $PM_{2.5}$ and OA during the monsoon season, and shows a substantial contribution to secondary OA (SOA) particularly over the lower IGP. We find that the OA contribution to $PM_{2.5}$ is significant in all four seasons (17-30%), with primary OA generally representing the larger fractional contribution. We find that the volatility distribution of SOA is driven mainly by the mean total OA loading and the washout of aerosols and gas-phase aerosol precursors that result in SOA being less volatile during the pre-monsoon and monsoon season than during the post-monsoon and winter seasons.

## 1 Introduction

The Indo-Gangetic Plain (IGP), including parts of Pakistan, India and Bangladesh (Figure 1), is one of the most populous and polluted areas in the world. **It is home to ∼ 700 million people (9% of the global population (Bangladesh Bureau of Statistics, 2011; Indian National Commission on Population, 2020; Pakistan Bureau of Statistics, 2017)) and to the**

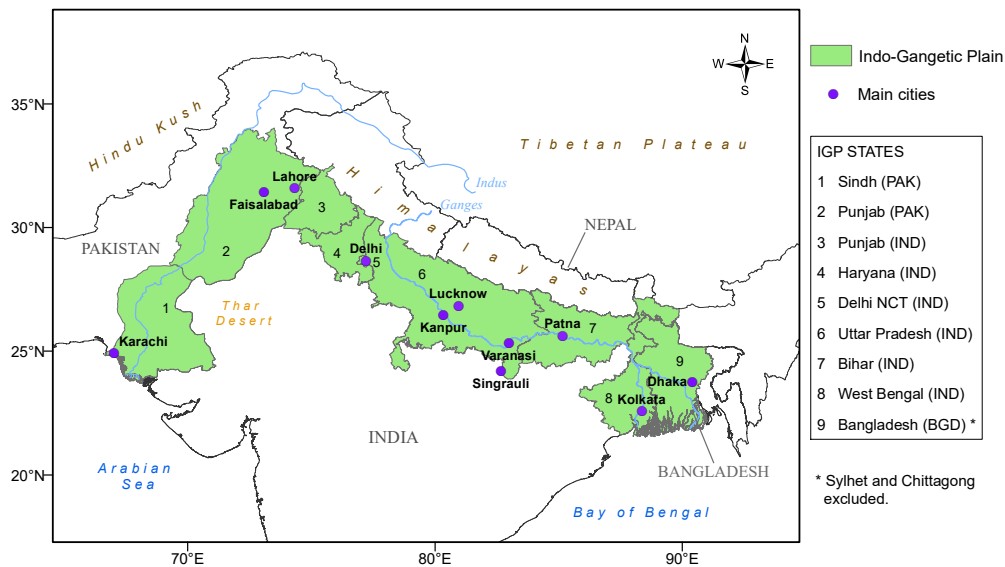

**Figure 1.** Geographical and administrative features of the Indo-Gangetic Plain (IGP), including Pakistan, India, and Bangladesh. Numbers denote individual IGP states and purple dots denote the main cities.

associated sources of anthropogenic air pollution, which are distributed proportionally to population, with hotspots
over cities of various sizes from megacities of more than 10 million people, e.g. Karachi, Lahore, Delhi, Kolkata, and
Dhaka, to smaller cities of a few million inhabitants, e.g. Faisalabad, Patna, Kanpur, Lucknow, and Varanasi (DESA,
2018). It has been estimated that there would be a potential gain in life expectancy in the IGP of approximately 4-6 years
if levels of $PM_{2.5}$ were reduced to standards set by the World Health Organisation (Greenstone et al., 2020; WHO, 2016).
The unique geography of the IGP and broader scale meteorological drivers, coupled with the regional diversity of seasonal
pollutant emission sources makes this region one of the most challenging places to study the controls of its air pollution and
the consequent impact on human health. Here, we use the WRF-Chem regional atmospheric chemistry and transport model to
describe the seasonal patterns of surface organic aerosol and $PM_{2.5}$ and to help disentangle the role of anthropogenic, pyrogenic
and biogenic emissions on their surface patterns across the IGP.

    The importance of the IGP lies in the fertility of its soils formed from alluvium that is deposited across the Indus and Ganges
basins by the Indus and Ganges rivers. These rivers originate in the Himalaya mountains and the Tibetan Plateau. The Indus and
Ganges basins benefit also from precipitation from the seasonal monsoon. The monsoon timing also defines the main seasons
over the IGP (India Meteorological Department): the pre-monsoon season runs from March to May, the monsoon season is

from June to September, the post-monsoon season is from October to December, and winter occurs in January and February. The Indian states across the IGP (e.g. Punjab, Haryana, and Uttar Pradesh) represent the vast majority of nationwide wheat and rice production. Rice and wheat are planted in May and November and harvested in October-November and April-May respectively, following the rice–wheat cropping cycle. The IGP is also an important producer of sugarcane, cultivated mainly in the Indus Valley in Pakistan and in the Indian state of Uttar-Pradesh. The two main seasons for planting are in September-October and February-March, followed by harvesting during the winter and pre-monsoon months, respectively. Crop residues left from harvesting, e.g. husk, bran, straw, are generally burned in open fires. Traditionally, these residues were ploughed back into the soil to maintain fertility and stability, but the sheer scale of current production precludes these practices in time for a second growing season (Chauhan et al., 2012; Ahmed et al., 2015). Open burning of these residues across the IGP, particularly during the post-monsoon season, is a large source of gaseous and particulate pollution that has implications for regional air quality and human health (Vadrevu et al., 2011; Jethva et al., 2019; Sembhi et al., 2020). Residential biofuel combustion also plays an important role for air quality (Conibear et al., 2020; Agarwala and Chandel, 2020).

The high population density and intense human activity over the IGP result in anthropogenic emissions being a major source of regional surface air pollution (Begum et al., 2013; Guttikunda and Jawahar, 2014; Shahid et al., 2015; Venkataraman et al., 2018). Residential energy consumption represents a major contribution to anthropogenic emissions with a large fraction of the rural and urban population using solid fuel for cooking (Conibear et al., 2018). Emissions from land transportation, particularly in cities, also represents a significant contribution to anthropogenic emissions (Begum et al., 2013; Guttikunda et al., 2014; Mallik and Lal, 2014). Intense agriculture over the IGP is associated with large emissions of ammonia, an aerosol precursor, from urea fertilizer application, as well as from post-harvest burning as described above (Kuttippurath et al., 2020; Wang et al., 2020). Vegetation cover over the IGP consists mainly of croplands (Stibig et al., 2007; Gumma et al., 2019), which have lower isoprene emissions than trees (Hardacre et al., 2013). Consequently, biogenic emissions over the IGP are lower compared to other parts of South Asia (Guenther et al., 2006; Stavrakou et al., 2014).

Regional dispersion of air pollution over the IGP is dominated on a seasonal timescale by the monsoon system, influenced by the high mountain ranges of Hindu Kush and Himalayas that lie to the northwest to northeast of the IGP. Agricultural planting and harvesting (and associated burning) are determined by the timing of the monsoon when the majority of the annual rainfall falls. Consequently, observed variations of $PM_{2.5}$ reflect large-scale variations in meteorology and the seasonal variations in anthropogenic, biogenic, and pyrogenic emissions (Jethva et al., 2005; Lelieveld et al., 2018; Schnell et al., 2018).

A growing body of regional models have been used to study the relationship between emissions, meteorology, and $PM_{2.5}$ over India (Kumar et al., 2015b; Bran and Srivastava, 2017; Kulkarni et al., 2020; Ojha et al., 2020), and to estimate the health impacts of outdoor exposure to $PM_{2.5}$ (Ghude et al., 2016; Conibear et al., 2018; David et al., 2019). Many studies have focused on post-monsoon biomass burning episodes and on air pollution during the winter season over the upper-central Indian part of the IGP (Guttikunda and Gurjar, 2012; Ram et al., 2012; Pant et al., 2015; Kumar et al., 2015a; Jethva et al., 2018; Singh et al., 2018; Krishna et al., 2019; Mhawish et al., 2020). But of course the IGP also includes parts of Pakistan and Bangladesh that remain poorly studied even though they are connected via atmospheric transport. With only a few exceptions, these studies have focused on total $PM_{2.5}$ although there is evidence that single aerosol components play a major role in $PM_{2.5}$ composition

over the IGP (Gani et al. (2019) and Singh et al. (2018) and references therein). Measurements have shown that organic aerosol (OA), originating from anthropogenic, pyrogenic, and biogenic emissions, constitute a significant fraction (20-35%) of $PM_{2.5}$

across the IGP especially during post-monsoon and winter seasons (Ram et al., 2008; Alam et al., 2014; Rajput et al., 2014; Behera and Sharma, 2015; Sharma et al., 2016). OA exists as a complex mixture, comprising of thousands of individual organic compounds, and it is made up of primary OA (POA), emitted directly to the atmosphere, and of secondary OA (SOA) formed by the condensation of organic vapours as they become progressively less volatile through oxidation (Seinfeld and Pandis, 2016; Donahue et al., 2006). Changes in OA volatility is key for the formation of SOA, and it is particularly sensitive to temperature,

ambient concentration of OA, and nitrogen oxide levels (Shrivastava et al., 2017). We take advantage of the Volatility Basis Set (VBS) model, which helps to describe succinctly the evolving volatility of OA through oxidative chemistry in the atmosphere (Donahue et al., 2006, 2012; Chuang and Donahue, 2016), described below. This method has been used successfully in a range of modelling studies (Lane et al., 2008b; Bergström et al., 2012; Ahmadov et al., 2012; Zhang et al., 2013; Zhao et al., 2016).

We use the WRF-Chem regional atmospheric chemistry model to characterise the seasonal and spatial distributions and composition of $PM_{2.5}$ and OA in light of synoptic meteorology and emission drivers over three sub-regions of the IGP, including relevant parts of Pakistan and Bangladesh. We use a 1-D VBS model to describe the evolution of OA and its influence on $PM_{2.5}$, described in section 2. In section 2, we also describe the *in situ* and satellite measurements we use to evaluate our model. In section 3, we describe the seasonal meteorology over the IGP, the seasonal distributions and composition of $PM_{2.5}$ and OA, and the seasonal distribution of SOA volatility. In section 3 we also use a perturbative approach to understand the

sensitivity of $PM_{2.5}$ constituent distributions to changes in anthropogenic, pyrogenic and biogenic emissions and to seasonal changes in the atmospheric environment. We conclude our study in section 4.

## 2   Data and Methods

Here, we describe the WRF-Chem model that we use to understand the influence of anthropogenic, pyrogenic, and biogenic emissions on the atmospheric distribution of $PM_{2.5}$ and OA over the IGP.

### 2.1   Weather Research and Forecasting model coupled with Chemistry

We use v.3.9.1.1 of the the Weather Research and Forecasting (WRF) model coupled with Chemistry (WRF-Chem) (Grell et al., 2005) to describe the emissions and atmospheric chemistry and transport associated with gas and aerosol phase compounds over the IGP during 2017 and 2018. WRF uses the Advanced Research WRF (ARW) dynamical solver to solve the fully compressible, non-hydrostatic Euler equations that describe atmospheric flow. These calculations are coupled with atmospheric chemistry so that that our $PM_{2.5}$ and OA calculations are consistent with the meteorology.

Our study domain is defined as 17°–40° N and 64°–97° E, encompassing the IGP at a horizontal spatial resolution of 20 km and using 33 vertical levels that span from the surface to 50 hPa ($\simeq$19 km). **For the description of terrain data for the domain (land use and soil categories) we use MODIS IGPB 21-category data at 30 arc-seconds resolution ($\sim$ 1 km) (Friedl et al., 2010).** To define our initial conditions and lateral boundary conditions, and for nudging (Newtonian relaxation),

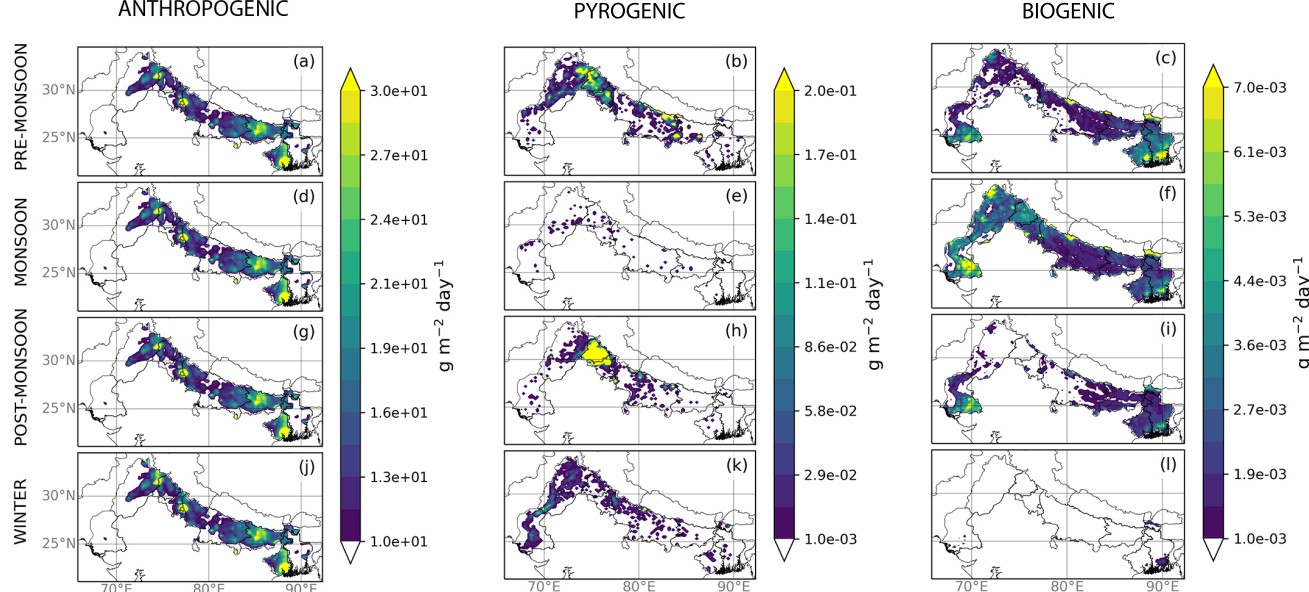

**Figure 2.** Seasonal mean daily emissions over the IGP (g m$^{-2}$ day$^{-1}$) of (left) column) anthropogenic, (middle column) biomass burning, and (right column) biogenic (isoprene) emissions. Anthropogenic emissions from EDGAR-HTAP and fire emissions from FINN. Biogenic emissions are calculated online in WRF-Chem using MEGAN. **To determine total anthropogenic and pyrogenic emissions we sum across all emitted species, respectively, while for biogenic emissions we consider only isoprene.**

we use meteorological reanalyses from NCEP FNL Operational Model Global Tropospheric Analyses Data (National Centers for Environmental Prediction, National Weather Service, NOAA, U.S. Department of Commerce, 2015) at a spatial resolution of $0.25° \times 0.25°$ and at a temporal resolution of six hours. We use the nudging approach at all levels to prevent our calculations from deviating too far from observed meteorology. Table B1 provides more details about the meteorological processes we use in our calculations. Chemical initial conditions and lateral boundary conditions for each month are provided by six-hourly CAM-CHEM global model data (Buchholz et al., 2019). We spin-up each simulation for a week before studying the model output to minimize the influence of the initial conditions.

To describe gas-phase chemistry we use the Model for OZone And Related chemical Tracers, version 4 (MOZART-4) chemical mechanism (Emmons et al., 2010), including the extended treatment of volatile organic compound (VOC) chemistry (Knote et al., 2014). Photolysis rates are calculated by the Fast Tropospheric Ultraviolet–Visible (FTUV) module (Tie et al., 2003).

We use the Model for Simulating Aerosol Interactions and Chemistry (MOSAIC) to simulate aerosols chemistry (Zaveri et al., 2008), including aqueous-phase chemistry (Knote et al., 2014). MOSAIC describes aerosols using four sectional discrete size bins: $0.039$—$0.156 \mu$ m, $0.156$—$0.625 \mu$ m, $0.625$—$2.5 \mu$ m, $2.5$–$10 \mu$ m. The first three of these bins represent PM$_{2.5}$, while the largest one describes coarse particulate matter (PM$_{2.5-10}$). We use the 1-D VBS method to describe SOA for WRF-Chem (Knote et al., 2015), based on previous studies (Lane et al., 2008b; Ahmadov et al., 2012). For each of the four aerosol size

bins in MOSAIC, the 1-D VBS implementation considers five volatility bins for semi-volatile organic compounds (SVOCs), described by effective saturation concentrations $C^*$ of $10^{-4}$, 1, 10, 100 and $10^3$ $\mu g\ m^{-3}$ at 298 K. The $log_{10}C^* = -4$ volatility corresponds to an inert compound, and serves computationally as a loss of particle phase organics to avoid unrealistic volatile mixtures due to continuously aging of gas-phase SVOCs. Lumped anthropogenic, pyrogenic, and biogenic gas-phase aerosol precursors undergo continuous gas-phase oxidation and partition between the gas and aerosol phase using pseudo-ideal partitioning theory (Pankow, 1994). Partitioning between the gas and aerosol phase depends on total organic aerosol load and temperature. SOA yields are also dependent on $NO_x$ levels, so SOA yields is calculated differently for low and high $NO_x$ conditions, through a branching ratio (Lane et al., 2008a). We also include the SOA formation from glyoxal (Knote et al., 2014). Loss of SVOCs is from washout via convective and grid scale precipitation. Our chosen implementation of VBS only accounts for SVOCs, and assumes that POA is inert so that it contributes only to the aerosol mass. We do not include direct emissions of SVOCs or intermediate VOCs (IVOCs). This is a limitation of our current implementation given evidence that SVOC and IVOC vapours creates a considerable amount of regional SOA, and that POA emissions are semivolatile and undergo oxidation and should be also considered in describing SOA production (Robinson et al., 2007). To describe POA using the VBS approach we would require information about the volatility distribution of POA, but conventional inventories typically consider POA as non-volatile. **The 1-D version of the VBS model is unable to describe some aspects of SOA formation, including fragmentation and the increase in OA oxidation state, which are better described by the 2-D version of the model that tracks the oxygen-to-carbon ratio (O:C) in addition to organic mass (Donahue et al., 2012). Previous studies have shown that the 2-D VBS model improves model-measurement agreement in SOA (e.g., Zhao et al. (2016)) but has a significant associated computational burden when used in 3-D chemistry transport models.** Further details of this VBS implementation in WRF-Chem are described in (Knote et al., 2015) and references therein.

We use monthly anthropogenic emissions from Emission Database for Global Atmospheric Research with Task Force on Hemispheric Transport of Air Pollution (EDGAR-HTAP v2.2) for year 2010 (Janssens-Maenhout et al., 2015) as provided by the WRF-Chem community, which provides the total anthropogenic emissions and includes a NMVOC speciation according to the gas and aerosol chemistry scheme we use here (MOZART-MOSAIC). **Using an anthropogenic emission inventory for 2010 to describe atmospheric chemistry during 2017-2018 will inevitably introduce some biases in our model PM$_{2.5}$ estimates, particularly because our study domain includes regions with rapidly growing emissions. From 2010 to 2017, India has seen reductions in BC, OC, CO and NMVOC emissions from the residential sector owing to policies that have enabled a switch to cleaner residential fuels and energy sources. However India's growing economy had led to a rapid increase of $NO_x$ and $SO_2$ emissions from the industrial sector($\sim +12\%$,$\sim +10\%$) and energy sector ($\sim +20\%$, $\sim +26\%$), and an increase in $NO_x$ and NMVOC from on-road transportation ($\sim +50\%$, $\sim +27\%$). An increase in intensive agricultural practices over the Indian IGP has increased ammonia emissions $NH_3$ ($\sim +15\%$) (McDuffie et al., 2020). Errors in PM precursor gaseous emissions will impact our ability to describe air pollution for our study year, especially for individual components of secondary inorganic aerosols (nitrate, sulfate and ammonium) and SOA. It remains difficult to disentangle the impact of using outdated emission estimates from other sources of model error, e.g., meteorology, chemistry, land-use change, and model resolution.** For pyrogenic emissions, hourly biomass burning emissions are taken from the

Fire Inventory from NCAR (FINNv.15) inventory for year 2017/2018 (Wiedinmyer et al., 2011). **Pyrogenic emissions are apportioned between FINN and EDGAR-HTAP inventories. The FINNv1.5 inventory includes global estimates of trace gas and particle emissions from open burning of biomass, which includes wildfire, agricultural fires, and prescribed burning (Wiedinmyer et al., 2011). EDGAR-HTAPv2.2 is focused on anthropogenic emissions but excludes large-scale biomass burning (e.g. forest fires, peat fires), agricultural waste or field burning. Within its residential sector, emissions include small-scale combustion, including heating, lighting, cooking and solid waste disposal or incineration (Janssens-Maenhout et al., 2015).** Biogenic emissions are calculated online using the Model of Emissions of Gases and Aerosol from Nature (MEGAN, Guenther et al. (2006)).

Figure 2 shows the seasonal distributions of total anthropogenic, pyrogenic, and biogenic (predominately isoprene) emissions over the IGP. Total anthropogenic emissions have been calculated by summing the mass contribution from all the chemical species (gas and particle) specified in the inventory once preprocessed onto the model domain using the WRF-Chem tools for the community ACOM-NCAR. We converted gas emissions to mass units using the appropriate molar mass for each species. The same approach has been used to calculate fire emissions, while isoprene emissions are calculated online by MEGAN in the WRF-Chem model and then converted to mass units. Anthropogenic emissions generally dominate in all seasons (Figure 2a,d,g,j) with daily values ranging from $10^1$ to $10^2$g m$^{-2}$ day$^{-1}$. The two largest localised regions of anthropogenic emissions are Delhi and Kolkata with emissions >100 g m$^{-2}$ day$^{-1}$, followed by smaller indian cities, e.g. Patna, Varanasi, Kanpur and Lucknow (Figure 1). Just south of the border of Uttar-Pradesh, the Madhya Pradesh district of Singrauli hosts several large power plants. The Pakistani and Bangladeshi parts of the IGP generally have the lowest anthropogenic emissions, with the exception of Karachi in south Pakistan, the north Pakistani Punjab (the most populated part of Pakistan where Lahore and **Faisalabad** are located), and Dhaka in Bangladesh. Emissions from Karachi and Dhaka have lower per capita emissions than Indian cities of comparable size.

Fires have a strong seasonal cycle, peaking during pre-monsoon and post-monsoon seasons (Figure 2b,h), with emissions $\sim 10^{-1}$g m$^{-2}$ day$^{-1}$ mainly due to agricultural stubble burning. The post-monsoon harvesting season includes fire emissions rates that are three times higher compared to the pre-monsoon season ($\sim 0.3$ g m$^{-2}$ day$^{-1}$ and $\sim 0.9$ g m$^{-2}$ day$^{-1}$, respectively). Post-monsoon fires are almost exclusively located in the Indian Punjab, with the largest values at the border with Haryana state. Pre-monsoon fires are located around the border of Pakistani and Indian Punjab and upper Haryana. There are also some isolated fires in the eastern part of the IGP. During winter (Figure 2k), low fire activity is present in the Indus valley in Pakistan and mainly over Uttar-Pradesh from post-harvesting of sugarcane crop.

Biogenic emissions peak during pre-monsoon and monsoon seasons (Figure 2c,f), with values of $2\times10^{-3}$g m$^{-2}$ day$^{-1}$ and $1.5\times10^{-2}$g m$^{-2}$ day$^{-1}$, respectively. The largest values are over Sindh in Pakistan, West Bengal, and Bangladesh. Land cover over the IGP is dominated by croplands, but state of Sindh includes coastal mangrove plantations, inlands riverine forests, irrigated plantations, and rangelands **(Ministry of Environment Government of Pakistan, 2009)**. Moreover, West Bengal and Bangladesh emissions are mostly confined close to the coast, where forest land is present (Reddy et al., 2016). During these two seasons there are also isoprene emissions over Uttar Pradesh from forests in Pilibhit and Kheri, and from northeast Pakistan.

For computational expediency we have chosen a representative period of one month for each distinct season over the IGP. We define, based on the seasonal definition of the Indian Meteorological Department (India Meteorological Department), the pre-monsoon period as 18th April to 16th May 2017; the monsoon season as 3rd to 31st July 2017; the post monsoon season at 18th October to 16th November 2017; and finally winter as 8th January to 5th February 2018. The 2017/2018 year is close to the climatological mean state so our results are typical of this region rather than being influenced by significant circulation changes due to, for example, El Niño Southern Oscillation climate variations (Null, 2020).

For the purposes of reporting our results we divide the IGP into three sub-regions: the upper IGP that includes the Pakistani states of Sindh and Punjab and the Indian Punjab; the middle IGP that includes the Indian states of Haryana, Delhi NCT, and Uttar Pradesh; and the lower IGP that include the Indian state of Bihar and West Bengal and Bangladesh, excluding the states of Chittagong and Sylhet (Figure 1).

## 2.2 Determining the Sensitivity of $PM_{2.5}$ and OA to Changes in Precursor Emissions

We use a perturbative approach to determine the importance of different source sectors on $PM_{2.5}$ and OA, which takes into account the non-linear chemical environment. Alternatively, setting a particular emission source to zero would result in a significant non-linear response that is unique to the source, consequently precluding any meaningful comparison of the importance of a particular source to $PM_{2.5}$ and OA.

First, we run a base run for each season. We then, for each season, systematically perturb one emission source by +5% over the study domain for the central week of each season, keeping the other sources the same as the base run. **Finally, we calculate the sensitivity $S_{ij}$ of species concentration to the changes in a given source of emissions as:**

$$S_{ij} = \frac{\Delta C_{ij}}{\Delta E} = \frac{\Delta C_{ij}}{E_{tot}^p - E_{tot}^b} = \frac{\sum_t (C_{ij,t}^p - C_{ij,t}^b)}{\sum_{ij,t,s} (E_{ij,t,s}^p - E_{ij,t,s}^b)}, \tag{1}$$

**where $\Delta C_{ij}$ represents the concentration change of our target species ($PM_{2.5}$ and OA in this study) at grid point $ij$ in response to an emission change $\Delta E$ summed over the IGP for a particular source. We perturb directly anthropogenic and fire emissions rates. Biogenic emissions are calculated online by scaling normalized emission rates by factors that describes changes in, for example, temperature, photosynthetic active radiation, leaf area index (LAI) (Guenther et al., 2006). We modify the WRF-Chem code to increment only isoprene emissions because our calculations suggest they account for almost all of biogenic emissions over the IGP, in agreement with other studies (Singh et al., 2011; Surl et al., 2018). $\Delta C_{ij}$ is calculated by summing over time the difference in concentrations at each grid cell ij of the perturbed run $p$ $C_{ij,t}^p$ and the base run $b$ $C_{ij,t}^b$. The change in concentration in each grid cell is therefore scaled by the same $\Delta E$, allowing to consider local and non-local emission influences equally and to avoid singularities in grid cells where there is no net emission change. We use this scaling because it allows us to compare the sensitivity of atmospheric concentrations to different sources types. $\Delta E$ is calculated as the difference of total emissions within the IGP domain between the perturbed model run and the base model run for a given source type.**

Total emissions across the IGP for the perturbed run $E_{tot}^p$ and for the base run $E_{tot}^b$ are calculated by summing emissions from all species for the length of the simulation and for all grid cells across the IGP. In more detail, emissions at

each grid point $ij$ for species $s$ between two consecutive model outputs at $t$ and $t+1$ is calculated (for both the perturbed and base runs) by $E_{ij,t,s} = \epsilon_{ij,t,s}\Delta t A_{ij}$. $\epsilon_{ij,t,s}$ denotes the emission rate of species $s$ at location $ij$ and output time $t$, $A_{ij}$ denotes the area of grid point $ij$, which in our calculations is constant at $400 \text{ km}^2$, and $\Delta t$ corresponds to an interval of model output which in our calculation is 3 hours. To take into account the different spatial variability of emissions from different sources (Figure 2), we scale $\Delta E$ with the total number of grid cells within the IGP for which the emission difference is $>0.001 \text{ g m}^{-2} \text{ day}^{-1}$, corresponding approximately to cumulative emissions $> 2.8$ Mg for each grid cell in one week. This threshold corresponds to a lower limit for significant emissions rate across the area considered (Figure 2). We also neglect values of $S_{ij}$ for which the change in the pollutant concentration $C_{ij}$ $<5\%$ of mean pollutant seasonal concentration over the IGP (4 $\mu\text{g m}^{-3}$ and 1 $\mu\text{g m}^{-3}$ for $PM_{2.5}$ and total OA, respectively). Using this additional threshold allows us to isolate significant changes in concentrations due to direct changes in emissions, and remove smaller values due to model non-linearity. We report the sensitivity parameter $S_{ij}$ with units of $\mu\text{g m}^{-3} \text{ Gg}^{-1}$. In a policy-making context, our sensitivity parameter provides information about how to control atmospheric concentrations by changing different emission sources in order to obtain the highest air quality benefits from certain emission reductions.

## 2.3 Data Used for Model Evaluation

We use *in situ* measurements of $PM_{2.5}$, $PM_{10}$, CO, $NO_2$, $O_3$, and $SO_2$ from the Indian Central Pollution Control Board CPCB and $PM_{2.5}$ data collected atop the US Embassy in Pakistan and Bangladesh U.S. Department of State. We accessed these data from the OpenAQ Platform (OpenAQ). Appendix B describes an overview of the *in situ* data, our data cleaning approach, and evaluation metrics. Given the lack of continuous measurements of OA and its **components** POA and SOA over the IGP, we compare our model OA with measurements available from the literature. We also evaluate the model using satellite observations of aerosol optical depth (AOD) from the NASA Moderate Resolution Imaging Spectroradiometer (MODIS) instrument aboard the Terra and Aqua satellites, which have a local equatorial overpass time of 1030 and 1330, respectively. AODs are retrieved at 550 nm, corresponding to particle sizes of $0.1–2\mu\text{m}$ and comparable to the $PM_{2.5}$ size range. In particular, we use the MODIS Collection 6.1 Level 2 combined Dark Target and Deep Blue AOD product available on a 10 km spatial resolution (Levy et al., 2013).

Here we summarize the main results of our evaluation (detailed results are available in Appendix B). We report the normalized mean bias (NMB) and the Pearson correlation coefficient $r$, which we use to describe how well the model reproduces the observations. The model tends to overestimate surface $PM_{2.5}$ concentrations (0.004<NMB<0.4) especially during monsoon season (NMB=0.4) but it has skill in reproducing observed seasonal variations ($r > 0.62$) with the exception of the monsoon season ($r = 0.09$). **Poorer model performance during the monsoon period may be due to a number of compounding factors. In particular, it is challenging to reproduce observed atmospheric water vapour and precipitation over the Bay of Bengal, western coasts of India and the Himalayan foothills during summer months. Uncertainties in the representation of topography, insufficient mixing in the boundary layer, errors in moisture transport and simulation of surface moisture availability, soil temperature and an excessive water vapour flux from the ocean all contribute to model error**

(Kumar et al., 2012a). Previous studies have shown that monsoonal rainfall is not well described by regional models such as MM5 or WRF (Rakesh et al., 2009; Ratnam and Kumar, 2005). When we compare our WRF model simulation with MERRA-2 reanalysed meteorology (Gelaro et al., 2017) we find that precipitation rates have a negative model bias of $\simeq 80\%$ over the IGP, similarly to what Conibear et al. (2018) obtained with a similar model set-up.

For $PM_{10}$, the model tends to underestimate the observation in all seasons (NMB up to -0.25) except in premonsoon season (NMB=0.15) and has poorer skill in reproducing observed $PM_{10}$ variations compared to $PM_{2.5}$ ($r \leq 0.69$), especially during winter and pre-monsoon season. **We generally find poorer model agreement with gas-phase pollutants, including a positive model bias and comparatively poor correlations with observations of $NO_2$, $SO_2$ $O_3$ (Table A3). We attribute this to multiple sources of error. Given the coarse spatial and temporal resolution our model (20 km$\times$20 km spatial, 3 hour temporal), we expect our model to be affected by non-negligible representation error due to the CPCB network sites often being located near to roadsides or in dense urban areas where the model will struggle to reproduce. This source of error preferentially affects reactive trace gases that react on timescales with transport across individual model grid cells. Previous studies have reported similar model limitations (Fountoukis et al., 2013; Paolella et al., 2018; Kuik et al., 2016; Tan et al., 2015; Sirithian and Thepanondh, 2016; Balasubramanian et al., 2020). Data for Pakistan are not available for our modelling study period (2017/2018) so we instead use data from 2019 for the monsoon and post-monsoon seasons and data from 2020 for the winter and pre-monsoon seasons, which represents an additional source of error. Previous studies show that regional modelling over south Asia tends to overestimate satellite column observations of $NO_2$ by 10–50% over the Indo-Gangetic Plain, the bias peaking as high at 90% during winter months (Kumar et al., 2012b), and up to +131% when compared to ground-based observations over densely populated urban regions (Karambelas et al., 2018). These differences have been attributed mainly to errors in $NO_x$ emission inventories over densely populated areas, uncertainties in seasonal variations of emissions, absence of diurnal and vertical profiles of anthropogenic emissions(Kumar et al., 2012b; Karambelas et al., 2018), and underestimation of precipitation rate that will reduce the loss of soluble trace gases Kumar et al. (2012a). Similarly, previous regional model studies of IGP region have tended to over-predict concentrations of $SO_2$, with NMB>3.5 (Conibear et al., 2018; Kota et al., 2018). We attribute our positive model bias of $SO_2$ to using an outdated emission inventory that does not take into account the beginning of a shift from coal to gas-based power plants (Sharma and Khare, 2017). Urbanisation has been shown to affect the diurnal spatial distribution of surface ozone (Li et al. (2014) and references therein), and also the magnitude and location of anthropogenic emissions of $NO_x$ and VOCs that subsequently affect surface ozone photochemistry (Zhang et al., 2004; Ghude et al., 2013). Finally, some fraction of the overestimation of surface ozone is linked to our use of the MOZART chemical mechanism that has been previously reported to have a positive model bias over south Asia compared to other mechanisms (Sharma et al., 2017). Collectively, these model limitations associated with describing reactive trace gases will impact our ability to model particulate matter, especially secondary components over urban areas across the IGP.**

For OA, the model reproduces the order-of-magnitude seasonal trends (Table B4) but additional measurements are needed to robustly assess model performance. Table B5 shows that WRF-Chem AOD agree with spatial distributions of MODIS AODs with $r$ typically $> 0.5$ with the exception of the monsoon season ($r$=0.35). **Poor model skill during the monsoon**

**season may reflect difficulties in retrieving AOD during extensive seasonal cloud coverage. In addition, the model has specific difficulties in reproducing atmospheric aerosol abundances during monsoon season, as highlighted earlier in this section, that could affect the simulated total AOD column.** The model tends to overestimate MODIS AOD during premonsoon (NMB=0.33, 0.26 for Terra and Aqua satellites) and slightly underestimate AOD in the other seasons (NMB ranges from -0.06 to -0.19).

## 3 Results

First, we summarise the seasonal meteorology over the IGP, which influences the physical and chemical environments that determine $PM_{2.5}$ and OA. We then report seasonal distributions of surface $PM_{2.5}$ and the corresponding constituent aerosol composition. Finally, we investigate the seasonal influence of POA and SOA on $PM_{2.5}$ and the volatility of the surface SOA across the IGP. In describing the seasonal distribution of $PM_{2.5}$, OA, POA and SOA we highlight the influence of anthropogenic, pyrogenic, and biogenic emissions and synoptic meteorology in shaping these patterns.

For the purpose of describing $PM_{2.5}$ and OA we begin our narrative with the post-monsoon season and finish with the monsoon season, reflecting the central importance of the monsoon system on atmospheric chemistry over the IGP. However, in the corresponding figures we retain the chronological order of events in a calendar year.

### 3.1 Seasonal Meteorological Drivers

Figure A1 shows model seasonal mean values for planetary boundary layer height (PBLH, m), surface relative humidity (RH, %), surface temperature (°C), mean daily rainfall (mm $day^{-1}$), and 10 m wind (m $s^{-1}$) over the IGP. Given that PBLH and RH show a diurnal cycle with high variance we report nighttime and daytime values for these variables.

During the pre-monsoon season, mean surface temperatures are higher than 30°C. Mean PBLH ranges from 1000 m up to 4500 m at daytime, with the highest values are over Pakistan and central IGP, and is almost an order of magnitude smaller during nighttime (120 m up to 400 m). Seasonal mean winds are typically 3 m $s^{-1}$, southward from the northern mountain chain of Hindu Kush and the Himalayas, and stronger northward from the coast, allowing pollutants to be transported mainly in the inland. Air is much more humid over the lowest part of the IGP (>60%). Rainfall follows similar patterns of RH, limited to Bangladesh with values below ∼3 mm $day^{-1}$.

During the monsoon season, the dominant feature is the monsoon itself. This manifests most obviously in increased rainfall, which increases the washout of hydrophillic pollutants, mainly in the central and lower part of the IGP, with mean daily rainfall values of 3-7 mm day$^{-1}$ with localized regions of rainfall in excess of 15 mm day$^{-1}$, and wind speeds in excess of 6 m s$^{-1}$ north-northeastward. Values of RH are >50% almost everywhere over the IGP, and relatively low values for the PBLH allow a well mixed chemical environment, with smaller day to night variation compared to pre-monsoon (1000-3000 m day, 500-1200 m night). Mean temperatures are similar to those during the pre-monsoon, with the most prominent increase over northern Pakistan (>35 °C).

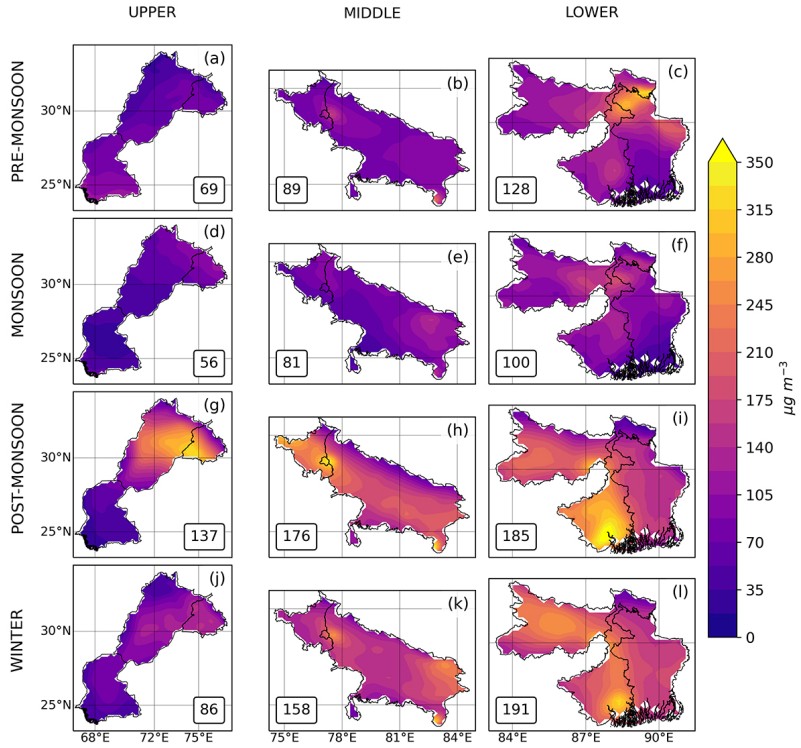

**Figure 3.** Seasonal mean spatial distributions of PM$_{2.5}$ ($\mu$g m$^{-3}$) over the upper, middle and lower IGP. The numbers inset of pre-monsoon (a–c), monsoon (d–f), post-monsoon (g–i), and winter (l–n) seasons denote the regional mean PM$_{2.5}$ value.

The post-monsoon season is characterized by cooler temperatures than the previous two seasons with mean values of $\sim$23°C, much lower values for PBLH (below 2000 m during day and $\sim$200 m during night), and weaker wind speeds (< 1 m s$^{-1}$ with no predominant direction, a combination of factors that results in pollution stagnation. With the exception of Bangladesh and the Indian states that are adjacent to the Bay of Bengal, rainfall is almost absent from the IGP. Nevertheless, air continues to be humid with the distribution and values of RH similar to the monsoon season, with values of up to 80% over the central and lower IGP, environmental conditions that favour water significantly contributing to PM mass without washout from rain.

During winter, mean temperature further drops to $\sim$15°C with cooler temperatures over regions adjacent to the northern mountain chains. PBLH values are at their daily annual minimum (<$\sim$1000 m) and its night values are similar to post-monsoon (<$\sim$200 m). Winds speeds are typically <12 m s$^{-1}$ with a net west-east gradient from the upper IGP to the lower IGP, which transports pollutants towards Bihar, West Bengal and Bangladesh, and with a north-south gradient over the Indus Basin that transports pollution from northern Pakistan to the coast. Daily rainfall is below 3 mm day$^{-1}$ anywhere across the IGP, but as for post-monsoon, RH remains high over the central and lower IGP (>40% daytime, 70% during nighttime).

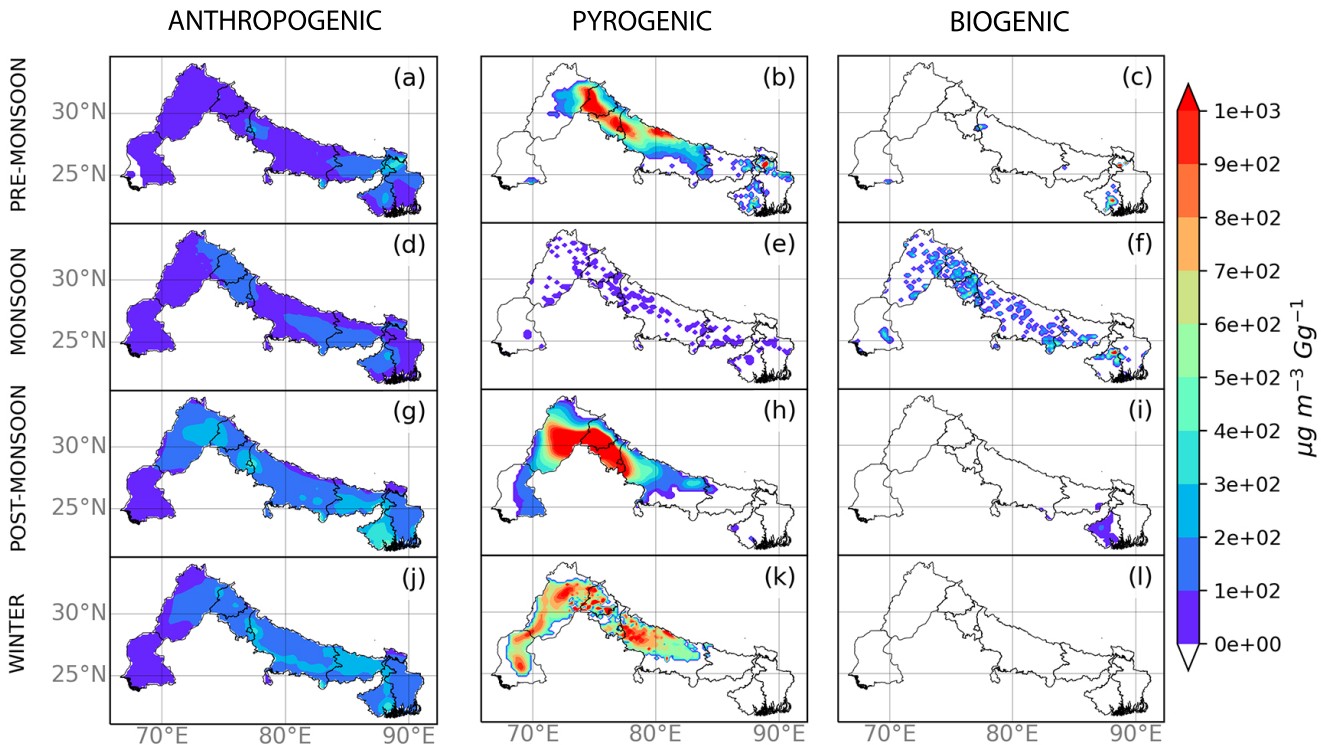

**Figure 4.** Seasonal sensitivity of PM$_{2.5}$ concentrations to changes in (left column) anthropogenic, (middle column) pyrogenic, and (right column) biogenic emissions ($\mu$ g m$^{-3}$Gg$^{-1}$). The calculation is described in the main text. Regions marked as white denote where sensitivity corresponds to PM$_{2.5}$ concentrations below the set threshold of 4 $\mu$g m$^{-3}$.

## 3.2 Seasonal Distributions of Surface PM$_{2.5}$

Figure 3 shows seasonal variations of surface PM$_{2.5}$ across the upper, middle, and lower IGP. Generally, we find the highest values of surface PM$_{2.5}$, up to 350 $\mu$g m$^{-3}$, during post-monsoon and winter seasons that are associated with lower PBLH allowing large anthropogenic emissions to accumulate in the boundary layer without ventilation from strong winds. From this section we begin **our** narrative from the post-monsoon season and finish with the monsoon season, but retain the figure panels in chronological order for a particular calendar year. Our seasonal distributions of PM$_{2.5}$ are similar to recent studies (Shahid et al., 2015; Ojha et al., 2020; Mhawish et al., 2020) although we report higher PM$_{2.5}$ concentrations especially over the lower IGP. **Compared to these studies, our model also takes into account water content in PM$_{2.5}$ mass in addition to dry PM$_{2.5}$ mass through aqueous phase chemistry. Our results shows that water content in PM$_{2.5}$ is substantial, especially over the lower IGP where water makes up to 42% of total PM$_{2.5}$ mass (see later in this section). This helps to explain our comparatively high PM$_{2.5}$ estimates.**

During the post-monsoon season (Figure 3 (g–i)), the mean values of surface $PM_{2.5}$ in the upper, middle, and lower IGP are 137 $\mu$g m$^{-3}$, 176 $\mu$g m$^{-3}$, and 185 $\mu$g m$^{-3}$, respectively. On a local scale, Kolkata and its surroundings in the lower IGP experience the worst air quality with mean $PM_{2.5}$ values in excess of 300 $\mu$g m$^{-3}$, closely followed by Delhi NCT, the border region between Indian and Pakistani Punjab, and Singrauli at the southern border of middle IGP ($\sim$300 $\mu$g m$^{-3}$). The best air quality is found in the Pakistani state of Sindh with $PM_{2.5}$ concentrations below 75 $\mu$g m$^{-3}$. Biomass burning in the Indian Punjab plays a key role in shaping the distribution of $PM_{2.5}$ during this season. Figure 4h shows that fire emissions have the largest impact on $PM_{2.5}$ concentrations across the Indian and Pakistani Punjab region, Haryana and Delhi NCT (sensitivities of up to $> 10^3$ $\mu$g m$^{-3}$ Gg$^{-1}$). **The impact of post-monsoon biomass burning emissions extends to the central part of the middle IGP over Uttar-Pradesh, where sensitivity of $PM_{2.5}$ to pyrogenic emissions (up to $6 \times 10^2$ $\mu$g m$^{-3}$ Gg$^{-1}$) is higher than anthopogenic emissions (up to $4 \times 10^2$ $\mu$g m$^{-3}$ Gg$^{-1}$).**

The sensitivity of $PM_{2.5}$ to changes in biogenic emissions (Figure 4i) have non-negligible values ($< 2 \times 10^2$ $\mu$g m$^{-3}$ $Gg^{-1}$) only over part of West Bengal in the lower IGP.

During the winter season (Figure 3(j–l)), wind patterns transport pollutants from the upper IGP to the lower IGP, resulting in west-east gradient in seasonal mean $PM_{2.5}$ concentrations . The mean $PM_{2.5}$ value in the lower IGP is 191 $\mu$g m$^{-3}$, the highest mean seasonal value for the IGP. The highest $PM_{2.5}$ concentrations are reached in Kolkata ($>$300 $\mu$g m$^{-3}$), and in the Bihar state, with a local peak in Patna ($>$220 $\mu$g m$^{-3}$) known as the 'Bihar pollution pool' (Kumar et al., 2018). In the middle IGP, mean $PM_{2.5}$ concentrations are 18 $\mu$g m$^{-3}$ lower than post-monsoon levels, with east Delhi and Singauli remaining the largest hotspots of the region ($>$220 $\mu$g m$^{-3}$). The upper IGP experiences the lowest seasonal $PM_{2.5}$ concentration (86 $\mu$g m$^{-3}$), lower than half the value in the lower IGP, with concentrations decreasing from the Punjab to the Sindth coast. Anthropogenic emissions dominate the distribution of $PM_{2.5}$ during winter over the lower IGP (sensitivity up to $4 \times 10^2$ $\mu$g m$^{-3}$ $Gg^{-1}$, Figure 4j), with the highest sensitivities over cities Kolkata, Singrauli. The influence of biomass burning is is significant over the the Indus basin, stretching until Uttar Pradesh (sensitivity up to $10^3$ $\mu$g m$^{-3}$ $Gg^{-1}$, Figure 4k), while biogenic emissions do not show a significant influence during this season (Figure 4l).

During the pre-monsoon season (Figure 3 a–c), air quality begins to improve due to higher PBLHs and stronger winds (Figure A1) that help to disperse pollutants. Mean $PM_{2.5}$ concentrations are similar over the upper and middle IGP with values lower than 90 $\mu$g m$^{-3}$. Higher concentrations remain in the lower IGP (128 $\mu$g m$^{-3}$) due to the accumulation of pollutants from the winds blowing from the Bay of Bengal to the slopes of the Himalayas over North Bangladesh. **High aerosol loading over the lower IGP during the premonsoon season is also influenced by biomass burning from Northeast India and Myanmar-Laos, which are partially included in our model domain.** $PM_{2.5}$ values over the upper part of the middle IGP (Figure3 b) show some influence from biomass burning (Figure 4 b). We find that anthropogenic emissions are most important over the lower IGP and localized region in the central IGP (Figure 4 a). $PM_{2.5}$ concentrations in Delhi NCT are jointly influenced by biomass burning and anthropogenic sources. Biogenic sources only have a significant impact over localized regions in the lower and middle IGP (Figure 4(c)).

Generally, the onset of the monsoon results in better air quality across the IGP due to higher rainfall rates, which increases wet deposition of aerosols, and higher PBLHs that improve the physical dispersal of surface emissions. Mean values of $PM_{2.5}$

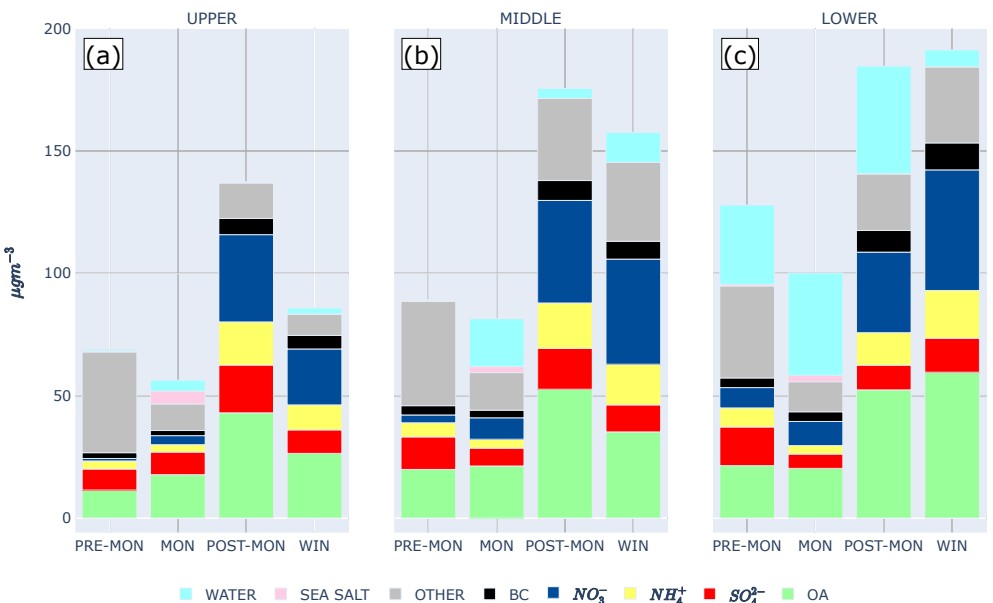

**Figure 5.** Seasonal mean PM$_{2.5}$ composition from the WRF-Chem model across the IGP: (a) upper, (b), middle, and (c) lower IGP. The constituents include sea salt (sum of sodium (Na) and chloride (Cl)), $NH_4^+$, $SO_4^{2-}$, $NO_3^-$, the sum of the remaining inorganic compounds (OTHER), total OA, BC, and liquid water.

are $\leq 100$ $\mu$g m$^{-3}$ across the IGP. The largest values of PM$_{2.5}$ are over the lower IGP (up to 170 $\mu$g m$^{-3}$). **We find that PM$_{2.5}$ is sensitive to biogenic emissions over localized regions across the IGP, where PM$_{2.5}$ can be more sensitive to changes in biogenic emissions than changes in anthropogenic emissions ($\sim$200-500 $\mu$g m$^{-3}$) and <200 $\mu$g m$^{-3}$, respectively).** Fires play only a small role in PM$_{2.5}$ during this season.

**Surface PM$_{2.5}$ composition**

**Figure 5 shows the modelled composition of PM$_{2.5}$ across the IGP.** Generally, we find more variability between seasons than across different parts of the IGP, except for the water contribution to PM$_{2.5}$ mass. The results we report for the chemical composition and seasonal trends of PM$_{2.5}$ are broadly consistent with chemical characterisation studies over the region (Chowdhury et al., 2007; Bhowmik et al., 2020). **As discussed in Section 2.3, model limitations in reproducing precursor trace gases will affect our ability to model secondary components of particulate matter. When comparing the model with recent field observations of PM$_1$ over Delhi during postmonsoon and winter (Gani et al., 2019; Gunthe et al., 2021; Patel et al., 2021), corresponding to two of our study seasons, we find that the model generally underestimates PM$_1$ (57-161 $\mu$g m$^{-3}$ observed, 17-22 $\mu$g m$^{-3}$ simulated) although we acknowledge that the model configuration we use is not ideal to model sub-micron PM due to our use of four sectional size bins. The model overestimates the contribution**

**of PM$_1$ from nitrate (6-11% observed, 11-13% simulated), but underestimates the contributions from sulfate (7-9% observed, 2% simulated) and organics (54-68% observed, 16-18% simulated).**

Inorganic species (secondary inorganic aerosol of sulfate, nitrate and ammonium and other inorganic aerosol) dominate the chemical composition by mass of PM$_{2.5}$, representing between 30–80% of total PM$_{2.5}$ for each season across the IGP. The mean seasonal mass of total inorganics across the IGP is 54–70 $\mu$g m$^{-3}$ during the pre-monsoon season, 27–35 $\mu$g m$^{-3}$ during the monsoon season,79–111 $\mu$g m$^{-3}$ during the post-monsoon season, and 51–114 $\mu$g m$^{-3}$ during winter. The largest inorganic aerosol values are found during the post-monsoon and winter seasons due to nitrate from fossil fuel combustion and from residential and energy use. We find a similar but relatively muted seasonal variation for black carbon with mass values between 2–11 $\mu$g m$^{-3}$. Sea salt transported from the coasts during the monsoon season adds 3–5 $\mu$g m$^{-3}$ (3–9%) to PM$_{2.5}$ across the IGP.

The water contribution to PM$_{2.5}$ is substantial over the lower IGP during pre-monsoon, monsoon, and post-monsoon seasons, with mass contribution of 32–44 $\mu$g m$^{-3}$ (25–42%), while during winter it accounts for 6 $\mu$g m$^{-3}$ (3.5%). For the middle IGP, water is a non-negligible fraction of PM$_{2.5}$ mainly during monsoon (20 $\mu$g m$^{-3}$, 24%) and winter (12 $\mu$g m$^{-3}$, 8%) seasons, while for the upper IGP the highest values of water mass are found during only the monsoon season (4 $\mu$g m$^{-3}$, 8%). The seasonal variation of water content reflects RH distritbuions, which above values of 60–70% allows PM hydrophilic components (e.g., nitrate, sulfate, sea salt) to uptake water via deliquescence.

The sum of primary and secondary OA contributes by mass between 17% and 31% of PM$_{2.5}$ across the IGP, with contributions from POA and SOA varying with season. During the pre-monsoon season, OA contributes 11–21 $\mu$g m$^{-3}$ to PM$_{2.5}$, representing 17–22% of the total mass. A similar mass contribution is found during the monsoon season (18–21 $\mu$g m$^{-3}$) but with higher percentage contribution to PM$_{2.5}$ (20–31%). The percentage mass contribution of OA to PM$_{2.5}$ is similar during the post-monsoon (28–31%, 43–52 $\mu$g m$^{-3}$) and winter (22–31%, 26–60 $\mu$g m$^{-3}$), with higher mass contribution during post-monsoon for the middle and lower IGP and during the winter season for the lower IGP. Our results for modeled PM$_{2.5}$ composition confirm the significance of OA contribution to fine particulate matter, and we analyse in more detail OA and its components in the next sections.

## 3.3   Seasonal Distribution of Surface OA

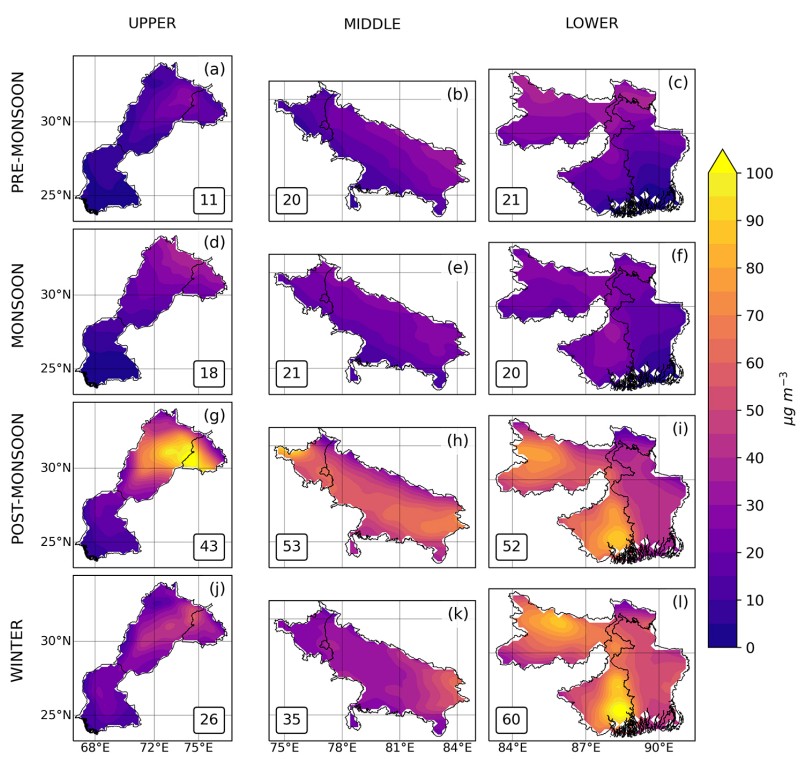

**Figure 6.** Seasonal mean distributions of total OA over the upper, middle and lower IGP. The numbers inset of pre-monsoon (a–c), monsoon (d–f), post-monsoon (g–i), and winter (l–n) seasons denote the regional mean total OA value.

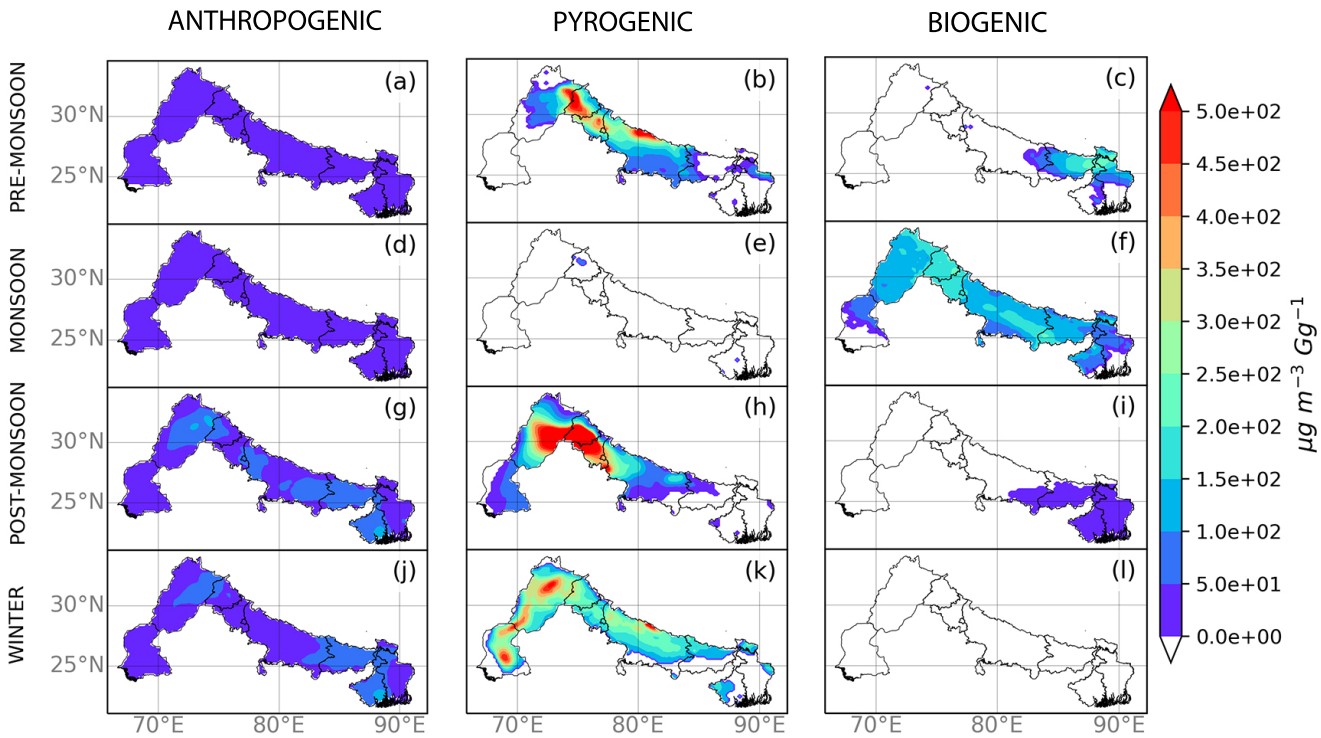

**Figure 7.** Seasonal sensitivity of total OA to changes in (left column) anthropogenic, (middle column) pyrogenic, and (right column) biogenic emissions ($\mu$ g m$^{-3}$Gg$^{-1}$). The sensitivity calculation is described in the main text. Regions marked as white shows where sensitivity corresponds to OA concentrations below the set threshold of 1 $\mu$g m$^{-3}$.

Figure 6 shows the season mean distributions of total OA with the corresponding POA and SOA distributions shown by Figures A2 and A3. We generally find that POA dominates seasonal values of total OA across the IGP, with the exception of the post-monsoon season when SOA and POA have comparable values.

During the post-monsoon season (Figure 6g-i), the largest OA concentrations are over the upper IGP at the border of Pak-
istani and Indian Punjab ($> 80$ $\mu$g m$^{-3}$), where POA values can exceed 50 $\mu$g m$^{-3}$. Although the largest regional mean is found over the lower IGP (52 $\mu$g m$^{-3}$) due to urban anthropogenic emissions in and around Kolkata and Patna where values are $>70$ $\mu$g m$^{-3}$. Over the middle IGP, the mean OA value is similar to the lower IGP (52 $\mu$g m$^{-3}$) but shows a more homogeneous distribution, with the highest OA values found at the borders between upper and lower IGP. Regional mean POA values range 23–29 $\mu$g m$^{-3}$ (Figure A2g-i), similar to SOA values (20–24 $\mu$g m$^{-3}$, Figure A3g-i). POA levels are much higher than
SOA over the Punjab states in India and Pakistan and in the Indian lower IGP ( 40–70 and 30–40 $\mu$g m$^{-3}$ for POA and SOA, respectively). Over the middle IGP, SOA is generally higher than POA (29 and 24 $\mu$g m$^{-3}$ for SOA and POA, respectively), with highest concentrations of SOA found in the lower Uttar Pradesh (up to 40 $\mu$g m$^{-3}$). Over Bangladesh and the Pakistani state of Sindth POA and SOA have comparable values ($<35$ $\mu$g m$^{-3}$).

**Similarly to PM$_{2.5}$, we find that during the post-monsoon season, the OA distribution across the IGP is most sensitive** to changes in biomass burning emissions (Figure 7g-i), with higher values over the Punjab to Delhi NCT, and part of Uttar Pradesh (up to $10^3$ $\mu$g m$^{-3}$ where fires are located over Indian Punjab). The sensitivity of OA to changes in biomass burning are localised, with POA most influenced by fires over Punjab and Haryana (Figure A4h) and corresponding impact on SOA extending over Pakistani Punjab and towards the middle IGP (Figure A5h). Similarly, biogenic emissions play only a localised role in OA and SOA concentrations where biogenic emissions are still significant during this season (Figures 7i) and A5i). OA are most sensitive to anthropogenic emissions over the Indian part of the lower IGP and in the **Pakistani Punjab** values (between 50-150 $\mu$g m$^{-3}$). We find that OA over the Delhi NCT megacity is not sensitive to these changes unlike other cities mentioned previously, so that Delhi is not one of the main hotspots of OA across IGP during this season (Figure 6h) unlike it is for PM$_{2.5}$ (Figure 3h). We find that the sensitivity of POA and SOA to changes in anthropogenic emissions are comparable across major cities of the Punjab states (Figures A4g, A5g).

We find that the largest seasonal mean values of OA are during winter over the lower IGP (60 $\mu$g m$^{-3}$, Figure 6j-l) with contributing localised peaks over Kolkata and Patna (>80 $\mu$g m$^{-3}$) and at the border between Pakistan and India (ranging 40–70 $\mu$g m$^{-3}$). Seasonal mean values of POA and SOA also peak during winter over the lower IGP (34 $\mu$g m$^{-3}$ and 26 $\mu$g m$^{-3}$, respectively.) During winter, the OA distribution is shaped by anthopogenic and pyrogenic emissions (Figure 7j-l). POA concentrations show to be sensitive to anthropogenic emissions in a similar way as it is for post-monsoon season (Figure A4g,j). SOA is also mostly determined by anthropogenic emissions over the lower IGP (Figure A5j). POA and SOA are also sensitive to pyrogenic emissions, but during this season it is limited to fires over the Indus basin in Pakistan and central IGP (Figures A4k, A5k). We find that biogenic emissions do not significantly influence OA during winter.

During pre-monsoon and monsoon seasons, the OA distributions (Figure 6a-f) have similar mean values over the middle and lower IGP (20–21 $\mu$g m$^{-3}$) and lower mean values over the upper IGP (11 and 18 $\mu$g m$^{-3}$, respectively). The highest POA concentrations are found at the border on India and Pakistan and over the lower IGP ($\simeq$30 and 40 $\mu$g m$^{-3}$, respectively). In both seasons, mean SOA concentrations are below 15 $\mu$g m$^{-3}$) across all the IGP. During pre-monsoon and monsoon seasons, OA concentrations are sensitive to anthropogenic emissions across the IGP with similar spatial distributions (Figure 7a,d). Pyrogenic emissions influence the OA distribution during the pre-monsoon season over the central IGP (Figure 7b), but OA less sensitive to these emissions compared with the post-monsoon season (Figure 7h). During the monsoon season, the influence of fires on OA is negligible across the IGP. The influence of biogenic emissions on OA, determined exclusively in our model via SOA, is limited to the lower IGP during the pre-monsoon season. During the monsoon season, these emissions have a widespread impact on OA (Figure 7f) with seasonal mean peak sensitivity of up to $2.3 \times 10^2$ $\mu$g m$^{-3}$ $Gg^{-1}$.

$PM_{2.5}$ **and OA are more sensitive to changes in biogenic emissions than changes in anthropogenic emissions during the monsoon period because of the role that anthropogenic emissions play in controlling the production of biogenic SOA. Previous studies have shown that anthropogenic emissions can enhance biogenic SOA production, with NO$_x$** concentrations playing a strong role in enhancing SOA formation from isoprene, and terpenes (Spracklen et al., 2011; Shilling et al., 2013; Shrivastava et al., 2019; Xu et al., 2020). A disadvantage of our using a single-variable perturbative

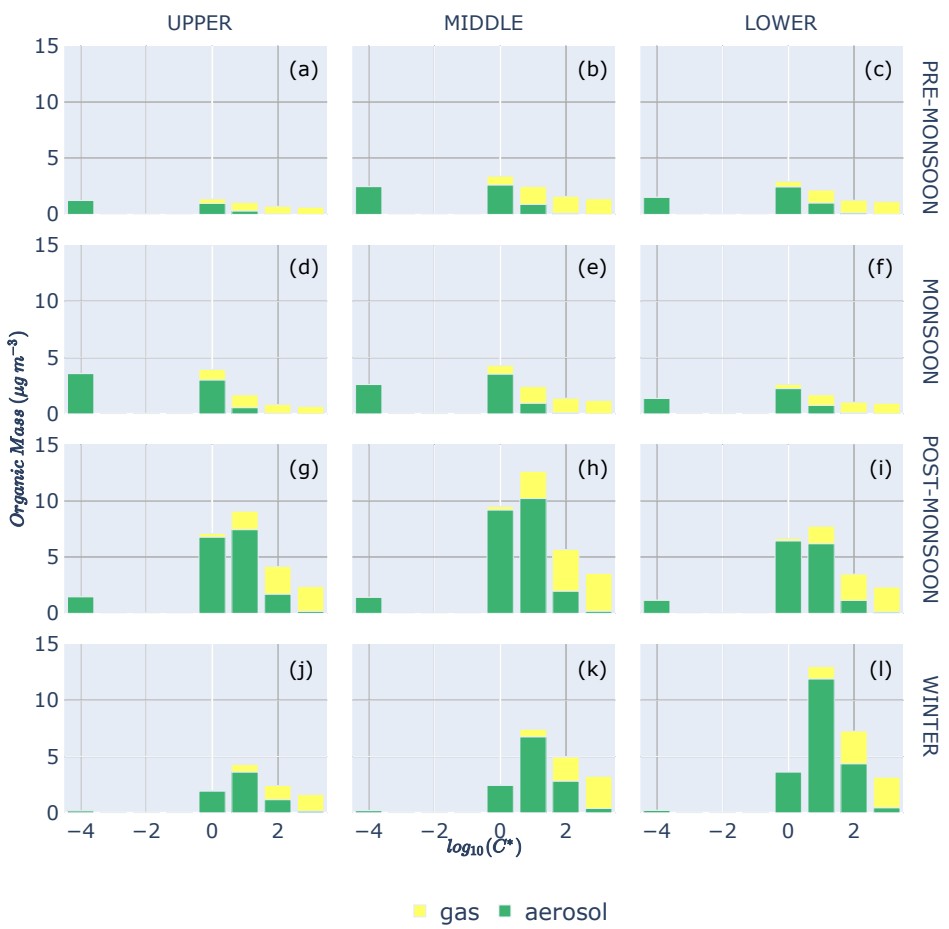

**Figure 8.** Seasonal mean volatility distribution of SOA over the upper, middle, and lower IGP as calculated within the WRF-Chem 1-D VBS scheme for (a–c) pre-monsoon, (d–f) monsoon, (g–i) post-monsoon, and (j–l) winter seasons.

**method is that we can only consider the impacts of one controlling factor in the production of OA. A study that considers the interactions between controlling factors is outside the scope of this study.**

### 3.4 Seasonal Distribution of SOA Volatility

We use aerosol volatility to describe how SOA is partitioned between the gas and particle phase to understand when it contributes to PM$_{2.5}$ mass loading. Figure 8 shows the seasonal mean volatility distributions for SOA across the IGP simulated using the 1-D VBS model in WRF-Chem (Knote et al., 2015). Seasonal and regional variations reflect changes in the physical and chemical environment in which the SOA is formed. Broadly, we find a gradual increase in the volatility of SOA from the

pre-monsoon season to the winter season, mainly reflecting the increase in the mean OA loading (Figure 6). Higher OA loading leads to a shift in the gas-particle partitioning towards more volatile bins, reflected in the seasonal variation in the population of the inert bin (denoted here as $log_{10}C^* = -4$, as described above). The contribution of this inert bin is negligible during winter and peaks during the monsoon season, with intermediate values during the pre-monsoon and post-monsoon transition seasons.

During the post-monsoon season, the particle phase organic mass is present at high volatility bins up to $log_{10}C^* = 2$. The largest particle phase mass loading (10 $\mu$g m$^{-3}$) is found over the middle IGP. The upper and lower IGP show a similar volatility distribution as the middle IGP but with lower mass loadings, with the lower IGP having the lowest mass loadings. The smallest values over the lower IGP reflect the persistence of rainfall over this region that leads to continued removal of water soluble gas-phase and aerosol-phase organics.

Surface-level atmospheric organic mass becomes even more volatile during the winter season, with particle phase organic matter present in all volatility bins. The largest mass loading for SOA are found over the lower IGP ($> 10$ $\mu$g m$^{-3}$) and decreases westwards towards the upper IGP, reflecting the E-W gradient of the total OA loading (Figure 6j–l).

SOA during the pre-monsoon (Figure 8 a–c) and monsoon (Figure 8 d–f) seasons are characterised by a volatility $\leq$ $log_{10}C^* = 1$, and with aerosol masses lower than 5 $\mu$g m$^{-3}$ for each volatility bin in both seasons. The higher volatility bins ($log_{10}C^* = 2$ and $\mathbf{log_{10}C^* = 3}$) are occupied exclusively by gas-phase organic compounds. We attribute this to water-soluble SVOCs being washed out by monsoonal rainfall. The washout of SVOCs results in gas-aerosol re-partitioning to establish thermodynamic equilibrium, associated with particle phase organics partitioning to the gas-phase. Aerosols are also removed via wet and dry deposition but we find most of the loss of SVOCs and SOA mass is lost via the gas phase (Knote et al., 2015). This also helps to explain the low levels of OA during the pre-monsoon and monsoon seasons (Figure 6). The OA volatility distribution is similar across the IGP, reflecting an approximately uniform physical environment during the two seasons (Figures A1 and 7).

## 4  Concluding Remarks

We used the WRF-Chem regional atmospheric chemistry model to understand the influence of anthropogenic, pyrogenic and biogenic emissions and meteorology on seasonal variations of the magnitude, distribution, and composition of PM$_{2.5}$ and organic aerosol across the Indo-Gangetic Plain (IGP) during 2017/2018.

**We find that the model reasonably reproduces concentrations of PM$_{2.5}$ in all seasons (NMB$<$0.2, r$>$0.6) except for the monsoon season (NMB=0.4, r=0.09), a reflection that modelling monsoonal meteorology remains challenging. However, uncertainty in our estimates remains on the individual PM$_{2.5}$ secondary components, given the limitation we found in the modeling to reproduce precursors gases surface concentrations when compared with observations. Availability of additional monitoring stations outside urban areas that are more representative of the spatial scales associated with model grid cells would help to evaluate model error, as well as use of finer-resolution and up to date inventories for precursors gases over the rapidly changing region of IGP.**

We find that the IGP experiences the highest seasonal mean levels of PM$_{2.5}$ during the post-monsoon (October—December, 166 $\mu g\ m^{-3}$) and winter (January—February, 145 $\mu g\ m^{-3}$) seasons with an heterogeneous distribution, in agreement with previous studies. The magnitude and distribution of anthropogenic emissions across the IGP are approximately constant throughout the year. During the post-monsoon season, agricultural burning emissions of post-harvest residues influence PM$_{2.5}$ mostly over the upper and middle IGP, particularly affecting the Indian and Pakistani Punjab region. These additional emissions are exacerbated by high pressure weather systems that reduce ventilation of surface air pollution to the free troposphere. During the winter season, ongoing anthropogenic emissions, wind patterns, and a seasonally shallow boundary layer result in a gradient in air quality from the upper to lower IGP, with the highest PM$_{2.5}$ values (in excess of 250 $\mu g\ m^{-3}$) over Kolkata and the state of Bihar. During the pre-monsoon (March–May) and monsoon (June–September) seasons wet scavenging of hydrophilic gas-phase aerosol precursors and aerosols, and more rigorous vertical mixing, reduces levels of PM$_{2.5}$ ( 95-79 $\mu g\ m^{-3}$ respectively).Generally, we find that PM$_{2.5}$ composition has a stronger seasonal variation than a geographical variation within each season. Total inorganic species dominate PM$_{2.5}$ composition (30-80%), with water uptake contributing substantially to the PM$_{2.5}$ mass especially over the lower IGP (up to 40%).

We find that OA represents a significant contribution to PM$_{2.5}$ throughout the year. On an annual mean basis, OA represents 17–30% of PM$_{2.5}$, with higher contributions during post-monsoon and winter seasons. Typically, POA contributes more to the OA loading than SOA in all seasons across the IGP. Anthropogenic and pyrogenic sources impact POA and SOA with similar patterns of PM$_{2.5}$ across the IGP during all seasons. Biogenic sources have a significant impact on SOA distribution across the IGP during the monsoon season but are limited to the lower IGP during the pre- and post- monsoon seasons. We find that the volatility distribution of SOA is driven mainly by the mean total OA loading and the washout of aerosols and gas-phase aerosol precursors that result in SOA being less volatile during the pre-monsoon and monsoon season than during the post-monsoon and winter seasons.

Mitigating levels of PM$_{2.5}$ over the IGP will require a range of regional and state-level policies that address the influences of intra- and inter- state anthropogenic, pyrogenic, and biogenic emissions. The relative influence of these emissions on PM$_{2.5}$ and the broader photochemical environment will likely change in the context of a warmer climate, e.g. biogenic emissions will increase as they are temperature dependent. It is therefore imperative that future studies should also consider sub-regional and city spatial scales, where individual sectors will be more important, and where there is the highest population density that will suffer from poor air quality.

## Appendix A:  Meteorological drivers, POA and SOA distribution

Figure A1 shows the mean seasonal WRF-Chem meteorological driver or pre-monsoon, monsoon and post-monsoon 2017 and winter 2018. Figure A2, to A5 show POA and SOA distribution over the IGP and their sensitivity to emissions drivers.

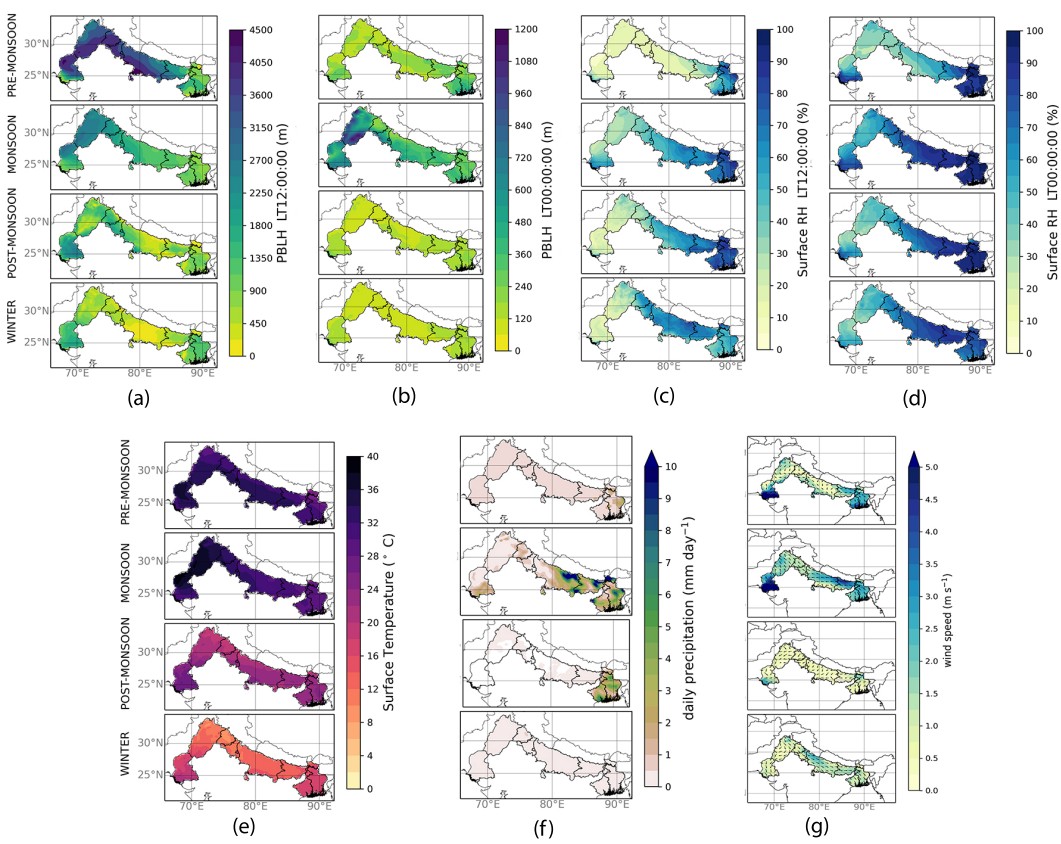

**Figure A1.** Seasonal mean WRF-Chem meteorological fields: (a) daytime planetary boundary layer height (m); (b) nighttime planetary boundary layer height (m); (c) daytime surface relative humidity (%); (d) nighttime surface relative humidity (%); (e) surface temperature at 2 m (°C; (f) daily precipitation rate (mm day$^{-1}$)); and (g) wind speed (m s$^{-1}$) and direction at 10 m.

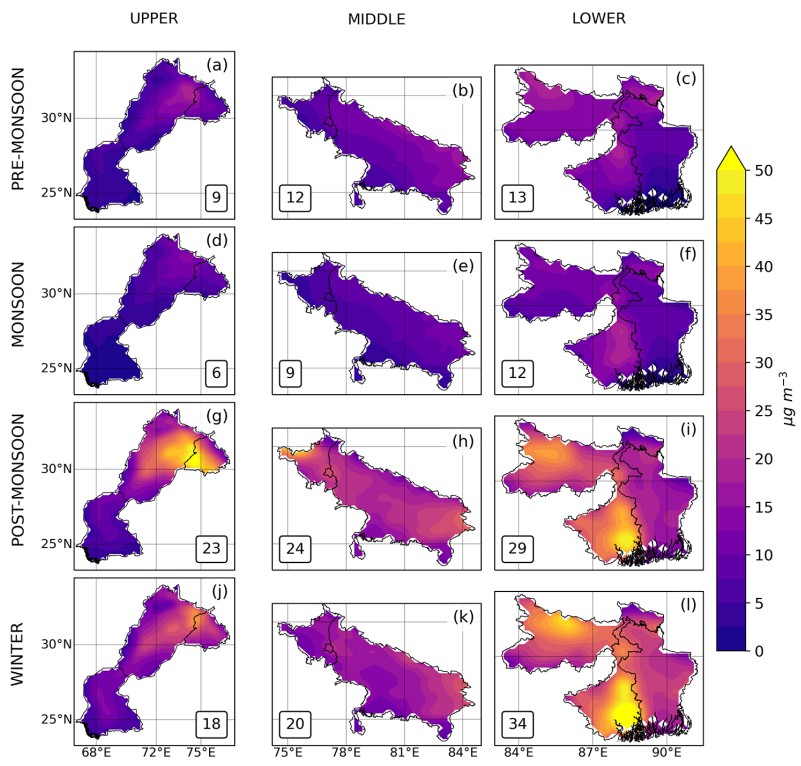

**Figure A2.** Seasonal mean distributions of POA over the upper, middle and lower IGP. The numbers inset of pre-monsoon (a–c), monsoon (d–f), post-monsoon g–i), and winter (j–l) seasons denote the regional mean POA value.

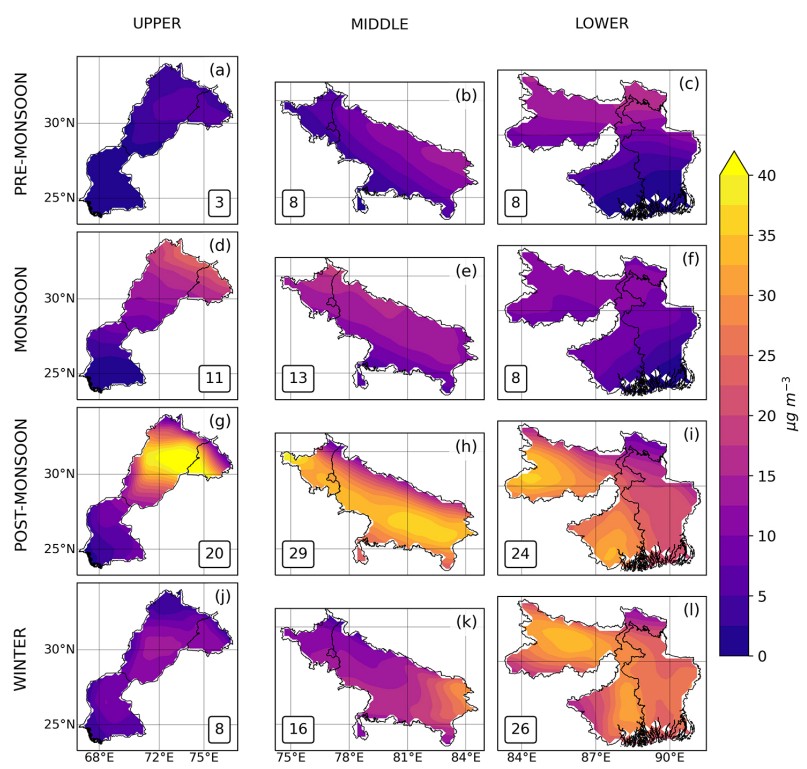

**Figure A3.** As A2 but for SOA.

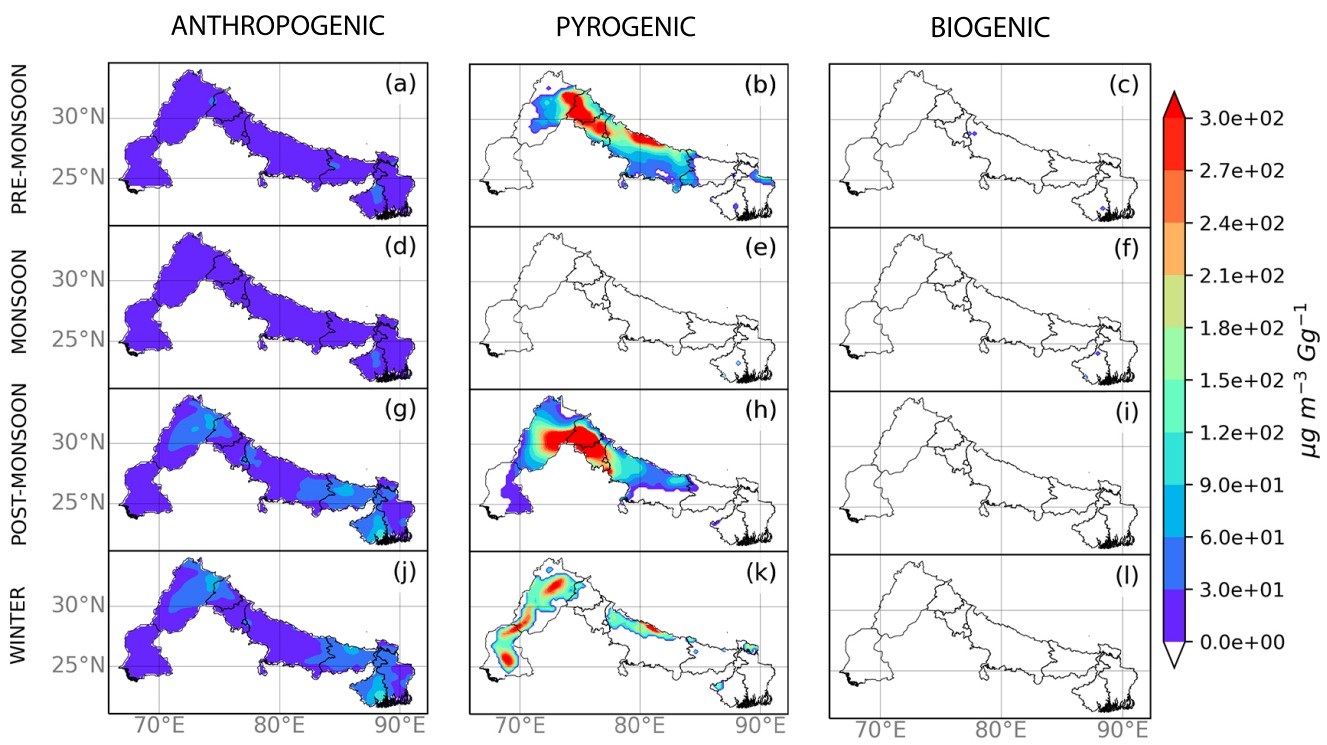

**Figure A4.** Seasonal sensitivity of POA to changes in (left column) anthropogenic, (middle column) pyrogenic, and (right column) biogenic emissions ($\mu$ g m$^{-3}$Gg$^{-1}$). The sensitivity calculation is described in the main text. Regions marked as white shows where sensitivity corresponds to OA concentrations below the set threshold of 1 $\mu$g m$^{-3}$.

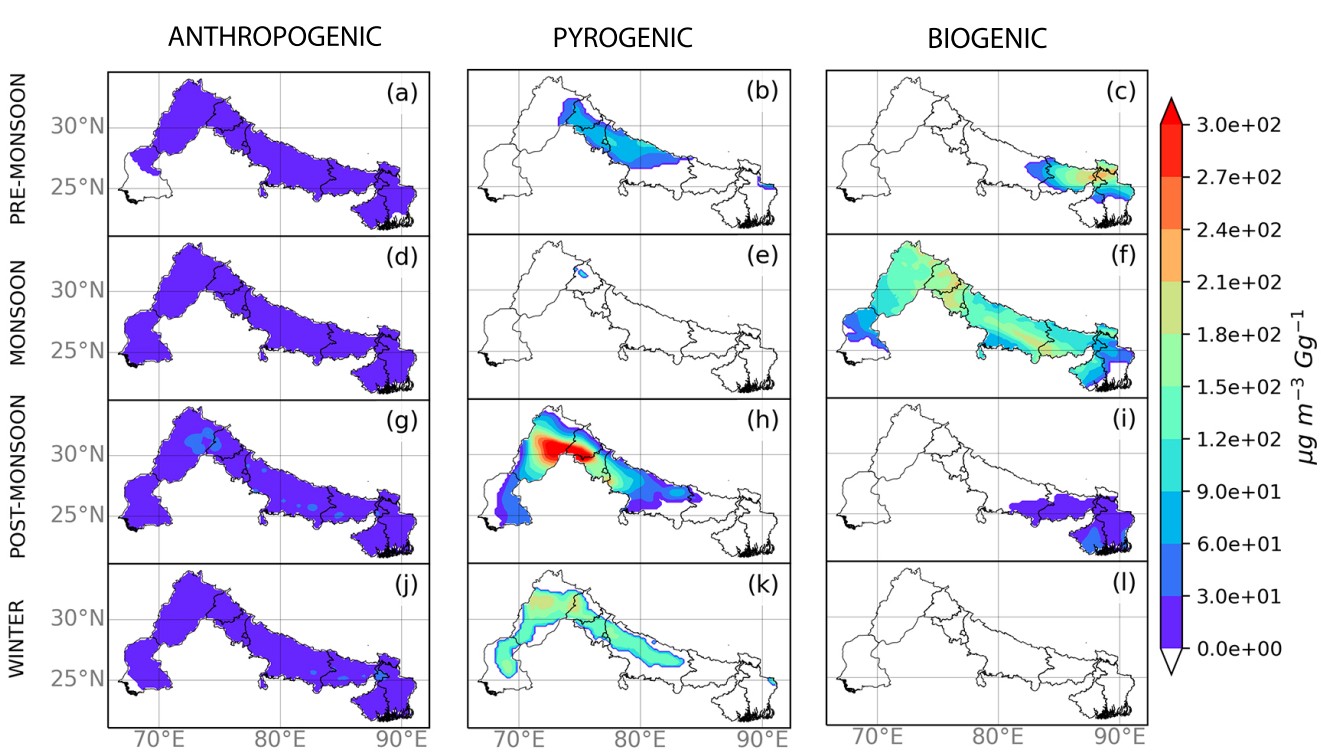

**Figure A5.** As Figure A4 but for SOA.

## Appendix B: WRF-Chem set-up and evaluation

Table B1 summarises the parametrisation for meteorology we use in WRF-Chem. The following subsections describe the evaluation of model result with ground based observation, OA values from literature and with satellite AOD respectively.

### B1    Ground-based Measurements evaluation

We use ground-based measurements from the Central Pollution Control Board and the U.S. Embassies, which are available for our 2017/2018 study period, and accessed through the OpenAQ Platform (OpenAQ). Data from Pakistan are only available from 2019 so we use 2019 data for the monsoon and post-monsoon seasons and data from 2020 for the winter and pre-monsoon seasons.

We apply a cleaning procedure of data for each pollutant. The cleaning procedure followed five sequential steps: 1) Exclude non valid, negative and zero values; 2) exclude hourly data with $z_{score} \geq 3$ respect to daily mean; 3) exclude days with fewer **than** 12 hourly measurements per day; 4) exclude stations with less then 15 days measurements per simulated season; and 5) exclude all stations but one if there are multiple stations in the same model grid-cell (for statistical independence in the comparison). From this cleaning procedure we get 31 independent stations (Table B2, **Figure B1**) with a total of seasonal measurements: 63 for CO, 54 for $SO_2$, 61 for $NO_2$, 50 for $O_3$, 84 for $PM_{2.5}$, 20 for $PM_{10}$. For particulate matter, we compare the dry mass of $PM_{2.5}$ and $PM_{10}$.

To compare the model against these measurements, we sample the model at the time and location of each measurement. In practice, we identify the model value closest to the measurement. We report seasonal mean statistics.

We evaluate the model using five metrics: the Mean Bias (MB), Root Mean Square Error (RMSE), Normalised Mean Bias (NMB), Mean Normalised Absolute Error (MNAE), and sample Pearson correlation coefficient ($r$). These metrics are widely used for air quality model evaluation (Zhang et al., 2006; Kumar et al., 2012b; Brasseur and Jacob, 2017; Conibear et al., 2018). Table B3 summarises the seasonal mean evaluation of the model with the metrics described.

### B2    Organic Aerosols

In the absence of continuous monitoring data of OA, we compare our model OC values with values found in the literature. Table B4 shows the comparison of modeled OC with measurements studies. **Location of measurement sites is shown in Figure B1** OA are converted from organic aerosol mass to organic carbon mass assuming OA/OC ratios 1.4 for POA and 2.0 for SOA, following (Knote et al., 2015).

### B3    Total AOD column

We compare our modeled prediction against satellite AOD retrievals **for 2017/2018** at 550 nm with a 10 km horizontal resolution obtained from both Terra (MOD04_L2) and Aqua (MYD04_L2) MODIS instruments. We use the best-quality AOD retrievals merged from the dark target and the deep blue algorithms (Levy et al., 2013). We re-grid the 10 km Terra and Aqua MODIS AOD data to the coarse WRF-Chem 20 km×20 km model grid.

We calculate the 550 nm AOD using WRF-Chem values at 300 nm and 1000 nm by interpolation using the Ångström power law. We sample the model at the local overpass time of Terra (1030) and Aqua (1330) where there exists at least one best-quality AOD retrieval. We then mean model and MODIS AOD values over time to generate seasonal statistics. Table B5 reports the main statistical metrics for AOD evaluation together with the range of observed and modeled AOD.

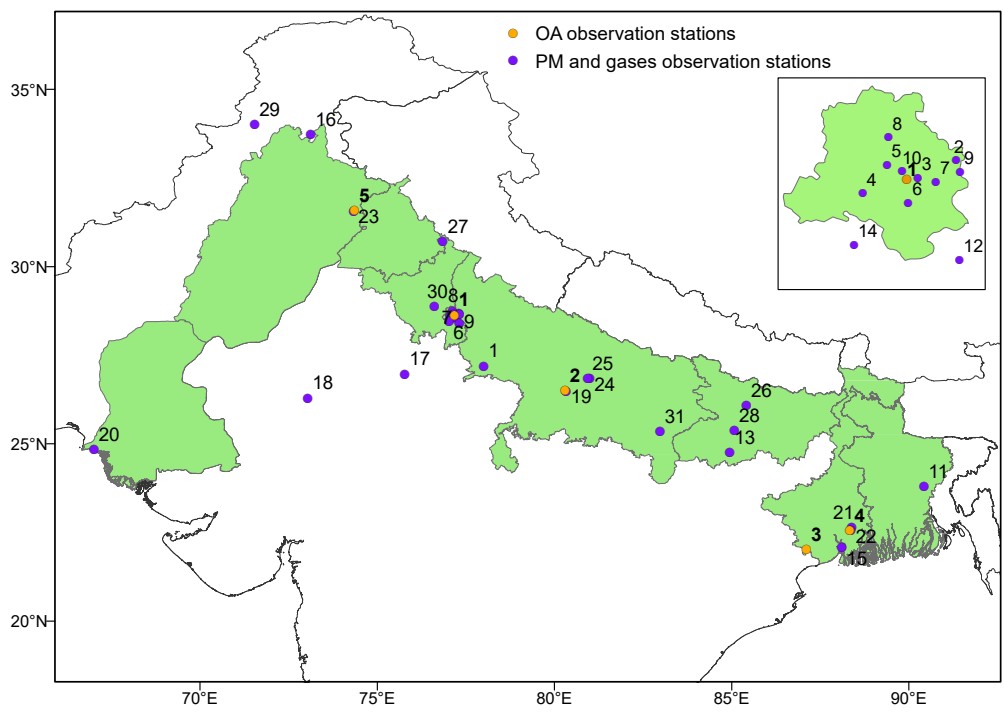

**Figure B1. Location of ground based observation for PM and gases (purple) and OA (orange). ID Number for each station correspond to ID number in table A2 and A4 respectively. The inset map shows in detail Delhi NCT.**

**Table B1.** Chosen parametrisations for meteorological processes in WRF-Chem.

| Process | Parametrisation |
|---|---|
| Cloud microphysics | Morrison double-moment scheme (Morrison et al., 2005) |
| Planetary boundary layer | Mellor-Yamada Nakanishi and Niino 2.5 (MYNN2) (Nakanishi and Niino, 2006) |
| Convection | Grell 3D scheme (Grell and Dévényi, 2002). |
| Short and long wave radiation | The Rapid Radiative Transfer Model (RRTM) (Iacono et al., 2008). |
| Land surface | Noah Land Surface Model coupled with a urban canopy model (Ek et al., 2003; Kusaka and Kimura, 2004) |

**Table B2.** List of selected ground-based stations and their measurements used for model evaluation.

| number ID | city | station name | latitude | longitude |
|---|---|---|---|---|
| 1 | Agra | Sanjay Palace | 27.20 | 78.01 |
| 2 | Delhi | Income Tax Office | 28.62 | 77.25 |
| 3 | Delhi | Delhi Technological University | 28.74 | 77.12 |
| 4 | Delhi | Shadipur | 28.65 | 77.16 |
| 5 | Delhi | Anand Vihar | 28.65 | 77.32 |
| 6 | Delhi | Punjabi Bagh | 28.67 | 77.12 |
| 7 | Delhi | NSIT Dwarka | 28.59 | 77.05 |
| 8 | Delhi | IHBAS | 28.68 | 77.31 |
| 9 | Delhi | Mandir Marg | 28.63 | 77.20 |
| 10 | Delhi | R K Puram | 28.56 | 77.17 |
| 11 | Dhaka | US Diplomatic Post: Dhaka | 23.80 | 90.42 |
| 12 | Faridabad | Sector16A Faridabad | 28.41 | 77.31 |
| 13 | Gaya | Collectorate - Gaya - BSPCB | 24.75 | 84.94 |
| 14 | Gurgaon | Vikas Sadan Gurgaon - HSPCB | 28.45 | 77.03 |
| 15 | Haldia | Haldia - WBPCB | 22.06 | 88.11 |
| 16 | Islamabad | US Diplomatic Post: Islamabad | 33.72 | 73.12 |
| 17 | Jaipur | VK Industrial Area Jaipur - RSPCB | 26.97 | 75.77 |
| 18 | Jodhpur | Collectorate Jodhpur - RSPCB | 26.29 | 73.04 |
| 19 | Kanpur | Nehru Nagar | 26.47 | 80.33 |
| 20 | Karachi | US Diplomatic Post: Karachi | 24.84 | 67.01 |
| 21 | Kolkata | US Diplomatic Post: Kolkata | 22.56 | 88.36 |
| 22 | Kolkata | Rabindra Bharati University, Kolkata - WBSPCB | 22.63 | 88.38 |
| 23 | Lahore | US Diplomatic Post:Lahore | 31.56 | 74.34 |
| 24 | Lucknow | Central School | 26.85 | 81.00 |
| 25 | Lucknow | Lalbagh, DN Park | 26.85 | 80.94 |
| 26 | Muzaffarpur | Collectorate - Muzaffarpur - BSPCB | 26.08 | 85.41 |
| 27 | Panchkula | Sector 6 Panchkula - HSPCB | 30.71 | 76.85 |
| 28 | Patna | IGSC Planetarium Complex - Patna - BSPCB | 25.36 | 85.08 |
| 29 | Peshawar | US Diplomatic Post: Peshawar | 34.01 | 71.54 |
| 30 | Rohtak | MD University, Rohtak - HSPCB | 28.88 | 76.62 |
| 31 | Varanasi | Ardhali Bazar | 25.35 | 82.98 |

**Table B3.** Statistical evaluation of model performance with ground based measurements for main PM and main gas pollutants.

| pollutant | season | NMB | NMAE | MB $[\mu g\ m^{-3}]$ | RMSE $[\mu g\ m^{-3}]$ | r |
|---|---|---|---|---|---|---|
| PM$_{2.5}$ | pre-monsoon | 0.12 | 0.31 | 10 | 34 | 0.62 |
| | monsoon | 0.41 | 0.65 | 19 | 36 | 0.09 |
| | post-monsoon | 0.19 | 0.28 | 33 | 62 | 0.84 |
| | winter | 0.004 | 0.28 | 0.7 | 53 | 0.69 |
| PM$_{10}$ | pre-monsoon | 0.15 | 0.59 | 32 | 133 | 0.11 |
| | monsoon | -0.21 | 0.28 | -25 | 46 | 0.69 |
| | post-monsoon | -0.14 | 0.36 | -41 | 122 | 0.66 |
| | winter | -0.25 | 0.43 | -64 | 131 | -0.85 |
| CO | pre-monsoon. | -0.64 | 0.64 | -643 | 825 | 0.44 |
| | monsoon | -0.55 | 0.55 | -428 | 567 | 0.12 |
| | post-monsoon | -0.65 | 0.65 | -1439 | 2272 | 0.29 |
| | winter | -0.52 | 0.61 | -703 | 1163 | -0.20 |
| NO$_2$ | pre-monsoon | 0.14 | 0.95 | 6 | 57 | 0.27 |
| | monsoon | 0.46 | 1.00 | 11 | 32 | 0.08 |
| | post-monsoon | 0.65 | 1.44 | 36 | 97 | 0.15 |
| | winter | 0.31 | 0.98 | 17 | 66 | 0.30 |
| O$_3$ | pre-monsoon | 1.59 | 1.67 | 75 | 91 | -0.52 |
| | monsoon | 2.92 | 2.92 | 64 | 66 | -0.12 |
| | post-monsoon | 2.96 | 2.98 | 98 | 113 | -0.75 |
| | winter | 2.87 | 2.92 | 71 | 87 | -0.55 |
| SO$_2$ | pre-monsoon | 0.25 | 0.85 | 3 | 13 | 0.04 |
| | monsoon | 0.27 | 1.38 | 4 | 34 | -0.18 |
| | post-monsoon | 2.36 | 2.44 | 33 | 49 | 0.51 |
| | winter | 1.85 | 2.15 | 27 | 43 | 0.04 |

**Table B4.** Comparison of modeled OC with measurements studies in the literature. Model values refers to the mean over the corresponding season of observations.

| number ID | location | period | species | OC obs [$\mu g\ m^{-3}$] | OC model [$\mu g\ m^{-3}$] | reference |
|:---:|:---:|:---:|:---:|:---:|:---:|:---:|
| 1 | Delhi | Jan-Feb 13-16 | PM$_{2.5}$ | 23.6 ± 12.9 | 23.4 | Jain et al. (2020) |
| | | Mar-May 13-16 | | 9.82 ± 4.16 | 13.3 | |
| | | Jun-Set 13-16 | | 6.77 ±2.63 | 12.1 | |
| | | Oct-Dec 13-16 | | 25.2 ±14.7 | 38.6 | |
| 1 | Delhi | Jan-Feb 13-16 | PM$_{10}$ | 30.1±12.1 | 23.5 | Jain et al. (2020) |
| | | Mar-May 13-16 | | 23.4 ± 10.7 | 13.4 | |
| | | Jun-Set 13-16 | | 15.9 ±9.7 | 12.3 | |
| | | Oct-Dec 13-16 | | 39.4 ±15.6 | 38.7 | |
| 2 | Kanpur | Oct-Nov 08 | PM$_{10}$ | 53.3 ±21.2 | 37.2 | Ram et al. (2012) |
| | | Dec 08 - Feb 09 | | 29 ± 14.5 | 21.9 | |
| | | Mar-Apr 09 | | 23.1 ± 11.5 | 12.3 | |
| 3 | Kharagpur | Nov 09 - Mar 10 | PM$_{2.5}$ | 30.7± 12.1 | 42.0 | Srinivas and Sarin (2014) |
| 4 | Kolkata | Jan-06 | PM$_{2.5}$ | 18.5±2.0 | 67.2 | Chatterjee et al. (2012) |
| | | Apr-May 06 | | 15.5± 3.6 | 7.8 | |
| | | Jul-06 | | 5±1 | 14.3 | |
| | | Oct-Nov 06 | | 11.5±5.0 | 57.4. | |
| 5 | Lahore | Jan-07 | PM$_{2.5}$ | 76.5 | 34.0 | Stone et al. (2010) |
| | | Apr-May-07 | | 43.5 | 20.0 | |
| | | Jul-07 | | 31.5 | 18.2 | |
| | | Oct-Nov 07 | | 111.2 | 61.0 | |

**Table B5.** Seasonal comparison of modeled total AOD column and satellite AOD observations for the Terra and Aqua instruments over the IGP **for the simulated period 2017/2018**.

| satellite | season | MB | NMB | NMAE | RMSE | r | range obs | range model |
|---|---|---|---|---|---|---|---|---|
| Terra | pre-monsoon | 0.33 | 0.53 | 0.56 | 0.44. | 0.53 | 0.06 - 1.78 | 0.10 - 2.29 |
| | monsoon | 0.04 | 0.05 | 0.45 | 0.52 | 0.35 | 0.00 - 3.42 | 0.10 - 3.78 |
| | post-monsoon | -0.05 | -0.06 | 0.25 | 0.25 | 0.76 | 0.07 - 3.50 | 0.12 - 1.71 |
| | winter | -0.11 | -0.19 | 0.30 | 0.21 | 0.64 | 0.05 - 1.74 | 0.11 - 1.16 |
| Aqua | pre-monsoon | 0.27 | 0.44 | 0.49 | 0.40 | 0.52 | 0.07 - 3.50 | 0.08 - 2.82 |
| | monsoon | -0.18 | -0.19 | 0.43 | 0.53. | 0.35 | 0.04 - 2.53 | 0.11 - 3.17 |
| | post-monsoon | -0.05 | -0.06 | 0.25 | 0.23 | 0.74 | 0.06 - 2.23 | 0.12 - 1.47 |
| | winter | -0.08 | -0.14 | 0.33 | 0.22 | 0.47 | 0.09 - 1.17 | 0.08 - 1.2 |

*Code and data availability.* All the data and materials used in this study are freely available. The WRF-Chem model code is available from https://www2.acom.ucar.edu/wrf-chem. NCEP FNL global tropospheric analyses were taken from https://rda.ucar.edu/datasets/ds083.3/. CAM-CHEM global model results were downloaded from https://www.acom.ucar.edu/cam-chem/cam-chem.shtml. The EDGAR-HTAPv2.2 emissions dataset ready for be used in WRF-Chem were downloaded from https://www2.acom.ucar.edu/wrf-chem/wrf-chem-tools-community. The FINN biomass burning emissions dataset was downloaded from https://bai.acom.ucar.edu/Data/fire/. Ground based observation used for the model evaluation where obtained from https://openaq.org/. The MODIS data are available from https://ladsweb.modaps.eosdis.nasa.gov/. Model setup files and code scripts for all the analysis described in the paper are available at DOI 10.5281/zenodo.5006024.

*Author contributions.* CM and PIP conceived the study and methodology. CM set-up the model with support from CK. CM performed the simulations, and led the model evaluation and data analysis, and interpreted the results together with PIP, CK and TJW. FY provided the AOD observations to be compared with the model AOD. CM and PIP wrote the paper, with input from all co-authors.

*Competing interests.* The authors declare that they have no conflict of interest.

*Acknowledgements.* CM is supported by funding from the Ford Motor Company University Research Program (#2016-2007R). The work of Fei Yao was supported by the China Scholarships Council/University of Edinburgh Scholarships. This study was funded as part of NERC's support of the National Centre for Earth Observation: PIP was supported by grant number #NE/R016518/1. We thank Jim Anderson, Wei Shen, and Sandy Winkler for helpful discussions. CM is thankful to the WRF-Chem community for the support received, in particular to S. Walters, G. Pfister, M. Barth, for providing support in the WRF-Chem online forum, and to L. Conibear, D. Lowe, R. Kumar, S. Archer-Nicholls, M. Morichetti for discussion and clarifications on specific technical issues. We acknowledge the use of the WRF-Chem preprocessor tool (anthro-emiss, fire-emiss, bioemiss, mozbc) provided by the Atmospheric Chemistry Observations and Modeling Lab (ACOM) of NCAR. We also acknowledge the use of the following software and packages for the post-processing and analysis of WRF-Chem simulations: ArcGIS 10.7, Python 3.6, xarray (Hoyer and Hamman, 2017), pandas (The pandas dev. Team, 2020) numPy (Harris et al., 2020) cartopy (Met Office, 2010 - 2015), plotly (Plotly Technologies Inc., 2015), matplotlib (Hunter, 2007), salem (Maussion et al., 2017). Administrative areas for IGP were taken from GADM (https://gadm.org). We thank two anonymous reviewers and the editor for their thoughtful comments on the manuscript.

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
