# Peer review of "Seasonal distribution and drivers of surface fine particulate matter and organic aerosol over the Indo-Gangetic Plain"

_Atmospheric Chemistry and Physics, 2021_

## Referee Comment (RC1)

**Reviewer Report of Mogno et al., 2021**

Mogno et al. present a comprehensive modelling study of seasonal distribution of particulate matter over the Indo Gangetic Plain (IGP), one of the world's most populated and polluted region. The manuscript is well-written and contains valuable results. This is a good overview paper on the drivers of air pollution in the IGP. However, the authors are requested to frankly acknowledge and openly discuss the large discrepancies between their simulation and observations, and additionally provide further description of their method of sensitivity analysis and its interpretaion.

1. Could you double check if the population of IGP is 400 million as per most recent estimates and perhaps provide a suitable reference?

2. L37: get rid of the extra "(June to September)"

- 3. Add a sentence on the limitations of using the 1D VBS scheme
- 4. L150: "Fasibalad" should be "Faisalabad"

5. Please confirm if emissions of all species have been added together to produce the plots in Fig 2.

- 6. L206: "components"
- 7. L270: "out" should be "our"

8. Emissions from 2010 have been used to simulate pollution in 2017-18. Comment on the nature of change in emissions that has occurred during this period (for example, you can consult McDuffie et al., 2020) and the expected changes in model results.

9. Provide your reasoning/explanation why your model results have reported higher PM2.5 levels over lower IGP

10. L281: You mention that "post-monsoon biomass burning do not impact the central and lower IGP". However, as per Figure 4h, if I read the colorscale properly, there's considerable influence of biomass burning (pyrogenic) emissions – on the scale of 250-500 ugm-3 Gg-1 all the way up to eastern Uttar Pradesh. These values appear to be higher than those from anthropogenic emissions in Figure 4g. Also, how do you reconcile this result with, for example, Ojha et al., 2020, who reported up to 20% contribution of biomass burning emissions to PM2.5 over central IGP cities? Since you are comparing values across different emission types, I suggest replotting Figure 4 and 7 using a common colorscale for all three emission categories.

11. Section 2.2: This section can benefit from a clearer description of the sensitivity analysis. For example, it is not clear what is  $\Delta t$  here – is it equal to a week, i.e., 7 days x 24 hours = 168 terms that were summed for each species for each gridpoint? Also, how is this particular sensitivity index supposed to be interpreted when thinking about control policies? It seems to me that you'd get a higher value for a grid if it is highly influenced by its own (local) emissions while other highly polluted grids may be influenced by emissions from several other grids leading to a much higher pollutant concentration but they won't show higher sensitivity values in this particular metric that you've

presented. This can cause confusion in interpreting such a metric especially for policymaking, when pollution is highly regional as is the case in IGP. Therefore, I request you to provide a richer discussion on the meaning and interpretability of the sensitivity index maps presented in Figure 4 and 7.

12. L352: "Punjabi Pakistan" would be better replaced by "Pakistani Punjab"!

13. Figure 8: Please increase size of the legend (gas,aerosol)

14. L400:  $log_{10}C^* = 2$  is written twice here. Should it be = 3 in the second instance?

15. L452: Replace "then" with "than".

16. Table A3: The model shows poor performance for NO2 all year round. Given that nitrate aerosol is a significant portion of PM2.5 as simulated by your modelling system particularly during post-monsoon and winter, how confident are you about this result? How does it compare with field observations such as Gani et al, 2019; Patel et al. 2021; Gunthe et al., 2021? Similary, how do you explain the dramatic overprediction of SO2 during post monsoon and winter and such low r-values? What are its implications for the simulated sulfate in your model? How do you explain the negative r-values for Ozone all year round? What are its implications for SOA simulation and regional contribution? These model weaknesses need to be acknowledged and discussed and the results and conclusions of the study need to be qualified in light of these weaknesses.

17. L351: You say that OA distribution during post-monsoon is most sensitive to changes in anthropogenic emissions but if I read the colorscales correctly, the sensitivity index is a maximum of 100 for anthropogenic emissions but is way above 100 for biomass burning emission over large parts of IGP. Again, please replot this figure with a common colorscale and reinterpret.

18. Figure 4 and 7: During the monsoon season, the sensitivity of  $PM_{2.5}$  and OA is higher towards changes in biogenic emissions than to changes in anthropogenic emissions. The reasons for this should be detailed in the text.

19. L299: Here you attribute the high PM2.5 concentrations in the lower IGP during pre-monsoon primarily to meteorology but this is actually a peak biomass burning season over the Myanmar-Laos region (see, for example, Figure 3 and 5 in Ansari et al., 2016; Figure 5 in Reddington et al., 2019). You can check your pyrogenic emissions file if it contains emissions beyond lower IGP, otherwise it must be coming from the chemical boundary conditions. You should include this influence in the sentence.

**References:**

Gani, S., Bhandari, S., Seraj, S., Wang, D. S., Patel, K., Soni, P., ... Apte, J. S. (2019). Submicron aerosol composition in the world's most polluted megacity: the Delhi Aerosol Supersite study. Atmospheric Chemistry and Physics, 19(10), 6843–6859. https://doi.org/10.5194/acp-19-6843-2019

Patel, K., Bhandari, S., Gani, S., Campmier, M. J., Kumar, P., Habib, G., … Hildebrandt Ruiz, L. (2021). Sources and Dynamics of Submicron Aerosol during the Autumn Onset of the Air Pollution Season in Delhi, India. ACS Earth and Space Chemistry, 5(1), 118–128. https://doi.org/10.1021/acsearthspacechem.0c00340

Gunthe, S.S., Liu, P., Panda, U. et al. Enhanced aerosol particle growth sustained by high continental chlorine emission in India. Nat. Geosci. 14, 77–84 (2021). https://doi.org/10.1038/s41561-020-00677-x

Ansari, T. U., Ojha, N., Chandrasekar, R., Balaji, C., Singh, N., & Gunthe, S.S. (2016). Competing impact of anthropogenic emissions and meteorology on the distribution of trace gases over Indian region. Journal of Atmospheric Chemistry, 73(4). https://doi.org/10.1007/s10874-016-9331-y

Reddington, C. L., Conibear, L., Knote, C., Silver, B. J., Li, Y. J., Chan, C. K., ... Spracklen, D. V. (2019). Exploring the impacts of anthropogenic emission sectors on PM\$\_{2.5}\$ and human health in South and East Asia. Atmospheric Chemistry and Physics, 19(18), 11887–11910. https://doi.org/10.5194/acp-19-11887-2019

McDuffie, E. E., Smith, S. J., O'Rourke, P., Tibrewal, K., Venkataraman, C., Marais, E. A., ... Martin, R. V. (2020). A global anthropogenic emission inventory of atmospheric pollutants from sector- and fuel-specific sources (1970--2017): an application of the Community Emissions Data System (CEDS). Earth System Science Data, 12(4), 3413–3442. https://doi.org/10.5194/essd-12-3413-2020

---

## Author Comment (AC1)

**Response to all referee comments (RCs) - MS No. acp-2021-69**

May 24, 2021

Many thanks to our referees for taking the time to give us thoughtful comments on our manuscript. Here we provide responses to these referee comments: referee comments are shown in bold, and our responses in plain text. The parts of the manuscript revised in response to comments are in italics here, and in bold in the revised manuscript.

**Response to Referee1 Comments (RC1)**

**Mogno et al. present a comprehensive modelling study of seasonal distribution of particulate matter over the Indo Gangetic Plain (IGP), one of the world's most populated and polluted region. The manuscript is well-written and contains valuable results. This is a good overview paper on the drivers of air pollution in the IGP. However, the authors are requested to frankly acknowledge and openly discuss the large discrepancies between their simulation and observations, and additionally provide further description of their method of sensitivity analysis and its interpretation.**

1. **Could you double check if the population of IGP is 400 million as per most recent estimates and perhaps provide a suitable reference?**

   Using the most recent estimates from national statistics department of Bangladesh (Bangladesh Bureau of Statistics, 2011) India (Indian National Commission on Population, 2020) and Pakistan (Pakistan Bureau of Statistics, 2017), the population in the IGP (as geographically defined in Figure 1) is estimated to be $\sim$ 700 million people, $\sim$9% of world population.

   We have amended the new estimate adding the references in the text:

   > Abstract: *The Indo-Gangetic Plain (IGP) is home to $\sim$9% of the global population.*

   > Introduction: *It is home to $\sim$ 700 million people (9% of the global population (Bangladesh Bureau of Statistics, 2011; Indian National Commission on Population, 2020; Pakistan Bureau of Statistics, 2017)).*

2. **L37: get rid of the extra "(June to September)"**
   Amended as suggested.

3. **Add a sentence on the limitations of using the 1D VBS scheme**

   We added the following sentence to the description of limitations of the 1D-VBS implementation used in our study:

   > Section 2.1: *The 1-D version of the VBS model is unable to describe some aspects of SOA formation, including fragmentation and the increase in OA oxidation state, which are better described by the 2-D version of the model that tracks the oxygen-to-carbon ratio (O:C) in addition to just organic mass (Donahue et al., 2012). Previous studies have shown that the 2-D VBS model improves model-measurement agreement in SOA (e.g., Zhao et al. (2016)) but has a significant associated computational burden when used in 3-D chemistry transport models.*

4. **L150: "Fasibalad" should be "Faisalabad"**
   Amended as suggested.

5. **Please confirm if emissions of all species have been added together to produce the plots in Fig 2.**

   Yes, emissions of all species are added together in Figure 2 for producing total anthropogenic and pyrogenic emissions, while for biogenic emissions only isoprene is considered, as indicated in the text (L139-144). For additional clarity, we have added the following sentence to Figure 2 caption:

   > Figure2 caption: *Emissions of all species are added together to produce total anthropogenic and pyrogenic emissions respectively, while for biogenic emissions we only consider isoprene.*

6. **L206: "components"**
   Amended as suggested.

7. **L270: "out" should be "our"**
   Amended as suggested.

8. **Emissions from 2010 have been used to simulate pollution in 2017-18. Comment on the nature of change in emissions that has occurred during this period (for example, you can consult McDuffie et al., 2020) and the expected changes in model results.**

   We highlight the changing in emissions between 2010 and 2017-18 and limitation of using a 2010 inventory for simulating pollution in 2017-18 as follow:

- Section 2.1: *Using an anthropogenic emission inventory for 2010 to describe atmospheric chemistry during 2017-2018 will inevitably introduce some biases in our model $PM_{2.5}$ estimates, since our study domain includes regions with rapidly growing emissions. From 2010 to 2017, India has seen reductions in BC, OC, CO and NMVOC emissions from the residential sector owing to policies that have enabled a switch to cleaner residential fuels and energy sources. However the increase in energy, industrial goods, and transport demand has led to a rapid increase of $NO_x$ and $SO_2$ emissions from the industrial sector($\sim +12\%$, $\sim +10\%$) and energy sector ($\sim +20\%$, $\sim +26\%$), and an increase in $NO_x$ and NMVOC from on-road transportation ($\sim +50\%$, $\sim +27\%$). Increase in intensive agricultural practices over the Indian IGP has increased ammonia emissions $NH_3$ ($\sim +15\%$) (McDuffie et al., 2020). Errors in PM precursor gaseous emissions will impact our ability to describe air pollution for our study year, especially for individual components of secondary inorganic aerosols (nitrate, sulfate and ammonium) and SOA. It remains difficult to disentangle the impact of using outdated emission estimates from other sources of model error, e.g., meteorology, chemistry, land-use model, and model resolution.*

9. **Provide your reasoning/explanation why your model results have reported higher PM2.5 levels over lower IGP**

Compared to previous estimates, our model set-up takes also into account water content in $PM_{2.5}$ mass in addition to dry $PM_{2.5}$ mass through aqueous phase chemistry simulation. Our model results shows that water content in $PM_{2.5}$ is substantial, especially over the lower IGP where it contributes up to 42% of total $PM_{2.5}$ mass (Figure 4). This at least helps to explain our higher $PM_{2.5}$ estimates compared to previous studies, especially over the lower IGP.

We added this explanation in the text:

- Section 3.2: *Compared to these studies, our model also takes into account water content in $PM_{2.5}$ mass in addition to dry $PM_{2.5}$ mass through aqueous phase chemistry. Our results shows that water content in $PM_{2.5}$ is substantial, especially over the lower IGP where water makes up to 42% of total $PM_{2.5}$ mass (see later in this section). This helps to explain our comparatively high $PM_{2.5}$ estimates.*

10. **L281: You mention that "post-monsoon biomass burning do not impact the central and lower IGP". However, as per Figure 4h, if I read the colorscale properly, there's considerable influence of biomass burning (pyrogenic) emissions – on the scale of 250-500 ugm-3 Gg-1 all the way up to eastern Uttar Pradesh. These values appear to be higher than those from anthropogenic emissions in Figure 4g. Also, how do you reconcile this result with, for example, Ojha et al., 2020, who reported up to 20% contribution of biomass burning emissions to PM2.5 over central IGP cities? Since you are comparing values across different emission types, I suggest replotting Figure 4 and 7 using a common colorscale for all three emission categories.**

We replotted Figure 4, 7 A4 and A5 (below) with a common colour scale. We changed also the colour palette to make changes of sensitivities clearer. Using a common scale, biomass burning emissions do indeed impact the central part of the middle IGP, as suggested by the reviewer. This result is consistent with Ojha et al. 2020. We reinterpret the figure as follows:

- Section 3.2: *The impact of post-monsoon biomass burning emissions extends to the central part of the middle IGP over Uttar-Pradesh, where sensitivity of $PM_{2.5}$ to pyrogenic emissions (up to $6 \times 10^2$ $\mu g\ m^{-3}\ Gg^{-1}$) is higher than anthropogenic emissions (up to $4 \times 10^2$ $\mu g\ m^{-3}\ Gg^{-1}$).*

[Figure]

Figure 4

[Figure]

Figure 7

[Figure]

Figure A4

[Figure]

Figure A5

11. **Section 2.2: This section can benefit from a clearer description of the sensitivity analysis. For example, it is not clear what is $\Delta t$ here – is it equal to a week, i.e., 7 days x 24 hours = 168 terms that were summed for each species for each gridpoint? Also, how is this particular sensitivity index supposed to be interpreted when thinking about control policies? It seems to me that you'd get a higher value for a grid if it is highly influenced by its own (local) emissions while other highly polluted grids may be influenced by emissions from several other grids leading to a much higher pollutant concentration but they won't show higher sensitivity values in this particular metric that you've presented. This can cause confusion in interpreting such a metric especially for policymaking, when pollution is highly regional as is the case in IGP. Therefore, I request you to provide a richer discussion on the meaning and interpretability of the sensitivity index maps presented in Figure 4 and 7.**

We modified the sensitivity method section as follows:

- Section 2.2: *Finally, we calculate the sensitivity $S_{ij}$ of species concentration to the changes in a given source of emissions as:*

$$S_{ij} = \frac{\Delta C_{ij}}{\Delta E} = \frac{\Delta C_{ij}}{E_{tot}^p - E_{tot}^b} = \frac{\sum_t (C_{ij,t}^p - C_{ij,t}^b)}{\sum_{ij,t,s} (E_{ij,t,s}^p - E_{ij,t,s}^b)}, \tag{1}$$

*$\Delta C_{ij}$ represents the concentration change of our target species ($PM_{2.5}$ and OA in this study) at grid point $ij$ in response to an emission change $\Delta E$ summed over the IGP for a particular source. We perturb directly anthropogenic and fire emissions rates. Biogenic emissions are calculated online by scaling normalized emission rates by factors that describes changes in, for example, temperature, photosynthetic active radiation, leaf area index (LAI) (Guenther et al., 2006). We modify the WRF-Chem code to increment only isoprene emissions because our calculations suggest they account for almost all of biogenic emissions over the IGP, in agreement with other studies (Singh et al., 2011; Surl et al., 2018). $\Delta C_{ij}$ is*

*calculated by summing over time the difference in concentrations at each grid cell ij of the perturbed run p $C_{ij,t}^p$ and the base run b $C_{ij,t}^b$. The change in concentration in each grid cell is therefore scaled by the same $\Delta E$, allowing to consider local and non-local emission influences equally and to avoid singularities in grid cells where there is no net emission change. We use this scaling because it allows us to compare the sensitivity of atmospheric concentrations to different sources types. $\Delta E$ is calculated as the difference of total emissions within the IGP domain between the perturbed model run and the base model run for a given source type.*

*Total emissions across the IGP for the perturbed run $E_{tot}^p$ and for the base run $E_{tot}^b$ are calculated by summing emissions from all species for the length of the simulation and for all grid cells across the IGP. In more detail, emissions at each grid point ij for species s between two consecutive model outputs at t and $t+1$ is calculated (for both the perturbed and base runs) by $E_{ij,t,s} = \epsilon_{ij,t,s}\Delta tA_{ij}$. $\epsilon_{ij,t,s}$ denotes the emission rate of species s at location ij and output time t, $A_{ij}$ denotes the area of grid point ij, which in our calculations is constant at 400 km², and $\Delta t$ corresponds to an interval of model output which in our calculation is 3 hours. To take into account the different spatial variability of emissions from different sources (Figure 2), we scale $\Delta E$ with the total number of grid cells within the IGP for which the emission difference is >0.001 g m⁻² day⁻¹, corresponding approximately to cumulative emissions > 2.8 Mg for each grid cell in one week. This threshold corresponds to a lower limit for significant emissions rate across the area considered (Figure 2). We also neglect values of $S_{ij}$ for which the change in the pollutant concentration $C_{ij}$ <5% of mean pollutant seasonal concentration over the IGP (4 μg m⁻³ and 1 μg m⁻³ for PM₂.₅ and total OA, respectively). Using this additional threshold allows us to isolate significant changes in concentrations due to direct changes in emissions, and remove smaller values due to model non-linearity. We report the sensitivity parameter $S_{ij}$ with units of μg m⁻³ Gg⁻¹. In a policy-making context, our sensitivity parameter provides information about how to control atmospheric concentrations by changing different emission sources in order to obtain the highest air quality benefits from certain emission reductions.*

12. **L352: "Punjabi Pakistan" would be better replaced by "Pakistani Punjab"!**
Amended as suggested.

13. **Figure 8: Please increase size of the legend (gas,aerosol)**
We increased the font size as suggested.

[Figure]

Figure8

14. **L400: log10C\* = 2 is written twice here. Should it be = 3 in the second instance?**
Yes, the second is should be =3. We amended as suggested.

15. **15. L452: Replace "then" with "than".**
Amended as suggested.

16. **Table A3: The model shows poor performance for NO2 all year round. Given that nitrate aerosol is a significant portion of PM2.5 as simulated by your modelling system particularly during post-monsoon and winter, how confident are you about this result? How does it compare with field observations such as Gani et al, 2019; Patel et al. 2021; Gunthe et al., 2021? Similarly, how do you explain the dramatic overprediction of SO2 during post monsoon and winter and such low r-values? What are its implications for the simulated sulfate in your model? How**

do you explain the negative r-values for Ozone all year round? What are its implications for SOA simulation and regional contribution? These model weaknesses need to be acknowledged and discussed and the results and conclusions of the study need to be qualified in light of these weaknesses.

We have revised the text in a few places to take into account this comment.

[revised manuscript text omitted]

17. **L351: You say that OA distribution during post-monsoon is most sensitive to changes in anthropogenic emissions but if I read the colorscales correctly, the sensitivity index is a maximum of 100 for anthropogenic emissions but is way above 100 for biomass burning emission over large parts of IGP. Again, please replot this figure with a common colourscale and reinterpret.**

We have now reinterpreted our results in light of the new plots that use a common colourscale:

- Section 3.3: *Similar to $PM_{2.5}$, we find that during the post-monsoon season, the OA distribution across the IGP is most sensitive to changes in biomass burning emissions (Figure 7g-i), with higher values over the Punjab to Delhi NCT, and part of Uttar Pradesh (up to $10^3$ $\mu g$ $m^{-3}$ where fires are located over Indian Punjab). The sensitivity of OA to changes in biomass burning are localised, with POA most influenced by fires over Punjab and Haryana (Figure A4h) and the corresponding impact on SOA extending over the Pakistani Punjab and towards the middle IGP (Figure A5h). Similarly, biogenic emissions play only a localised role in OA and SOA concentrations where biogenic emissions are still significant during this season (Figures 7i and A5i).*

18. **Figure 4 and 7: During the monsoon season, the sensitivity of PM2.5 and OA is higher towards changes in biogenic emissions than to changes in anthropogenic emissions. The reasons for this**

**should be detailed in the text.**

We have modified the text as follows:

- Section 3.3: *$PM_{2.5}$ and OA is more sensitive to changes in biogenic emissions than to changes in anthropogenic emissions during the monsoon period because of the role that anthropogenic emissions play in controlling the production of biogenic SOA. Previous studies have shown that anthropogenic emissions can enhance biogenic SOA production, with $NO_x$ concentrations playing a strong role in enhancing SOA formation from isoprene, and terpenes (Spracklen et al., 2011; Shilling et al., 2013; Shrivastava et al., 2019; Xu et al., 2020). A disadvantage of our using a single-variable perturbative method is that we can only consider the impacts of one controlling factor in the production of OA. A study that considers the interactions between controlling factors is outside the scope of this study.*

19. **19. L299: Here you attribute the high PM2.5 concentrations in the lower IGP during pre-monsoon primarily to meteorology but this is actually a peak biomass burning season over the Myanmar-Laos region (see, for example, Figure 3 and 5 in Ansari et al., 2016; Figure 5 in Reddington et al., 2019). You can check your pyrogenic emissions file if it contains emissions beyond lower IGP, otherwise it must be coming from the chemical boundary conditions. You should include this influence in the sentence.**

The file with pyrogenic emissions from pre-monsoon shows indeed biomass burning in the Northeast India and Myanmar:

[Figure]

Fire emissions in premonsoon season over model domain.

We include the following sentence:

- Section 3.2: *High aerosol loading over the lower IGP during the premonsoon season is also influenced by biomass burning from Northeast India and Myanmar-Laos, which are partially included in our model domain.*

**Response to Referee2 Comments (RC2)**

The authors use the WRF-Chem model to study the influences on fine particulate matter and organic aerosol (OA) over the Indo-Gangetic Plain (IGP). This paper presents a well constructed and informative sensitivity study to establish the extent to which PM2.5 concentrations are dependent on the strength of various emission sources, in different seasons. The paper is well written and certainly within the scope of ACP – I would recommend publication once the following minor issues are addressed.**General comments:**

1. **The distinction between anthropogenic and pyrogenic is a tricky one in reality so it would be useful to describe explicitly what is included in each category here. Is pyrogenic everything in the FINN inventory? For example, where do emissions from solid fuel combustion or agricultural burning fall? It's difficult (or impossible) to disentangle these two sources completely but it will be helpful for those doing further work in this area if you can clarify exactly which emissions are where.**

Thanks. This is an excellent point. We have now clearly defined what we mean by both terms.

- Section 2.1: *Pyrogenic emissions are apportioned between FINN and EDGAR-HTAP inventories. The FINNv1.5 inventory includes global estimates of trace gas and particle emissions from open burning of biomass, which includes wildfire, agricultural fires, and prescribed burning (Wiedinmyer et al., 2011). EDGAR-HTAPv2.2 is focused on anthropogenic emissions but excludes large-scale biomass burning (e.g.*

*forest fires, peat fires), agricultural waste or field burning. Within its residential sector, emissions include small-scale combustion, including heating, lighting, cooking and solid waste disposal or incineration (Janssens-Maenhout et al., 2015).*

2. **In general, the model performs quite poorly at simulating atmospheric composition during the monsoon season which may be related to issues with the precipitation or circulation patterns, rather than emission sources. I appreciate that these are beyond the scope of this study to explore but some further discussion of the performance of the model in this regard, or reference to studies that have looked at this, would be helpful to the reader.**

We amend as following the evaluation section:

- Section 2.3: *Poorer model performance during the monsoon period may be due to a number of compounding factors. In particular, it is challenging to reproduce observed atmospheric water vapour and precipitation over the Bay of Bengal, western coasts of India and the Himalayan foothills during summer months. Uncertainties in the representation of topography, insufficient mixing in the boundary layer, errors in moisture transport and simulation of surface moisture availability, soil temperature and an excessive water vapor flux from the ocean all contribute to model error (Kumar et al., 2012a). Previous studies have shown that monsoonal rainfall is not well described by regional models such as MM5 or WRF (Rakesh et al., 2009; Ratnam and Kumar, 2005). When we compare our WRF model simulation with MERRA-2 reanalysed meteorology (Gelaro et al., 2017) we find that precipitation rates have a negative model bias of $\simeq 80\%$ over the IGP, similarly to what Conibear et al. (2018) obtained with a similar model set-up.*

3. **Could you review the color scales for the figures? On several of them (e.g., Figure 2 and Figure A1 (f)) it's quite difficult to discern the variation that is referred to in the text because of the choice of values.**

Color scales for figure 2 and Figure A1 (f) and (g) have been revised to make the variations clearer:

[Figure]

Figure8

[Figure]

FigureA1

**Minor / specific comments:**

4. **4 Page 1, line 23: could you rephrase this description of the sources of pollution to clarify how they are distributed? If the pollution is concentrated over the cities, then you could say that it is distributed proportionally, according to population?**

   We rephrase the description as follow:

   - Introduction: *It is home to ∼ 700 million people (9% of the global population, (Bangladesh Bureau of Statistics, 2011; Indian National Commission on Population, 2020; Pakistan Bureau of Statistics, 2017)) and to the associated sources of anthropogenic air pollution, which are distributed proportionally to population, with main hotspots over cities of various sizes from megacities of more than 10 million people, e.g. Karachi, Lahore, Delhi, Kolkata, and Dhaka, to smaller cities of a few million inhabitants, e.g. Faisalabad, Patna, Kanpur, Lucknow, and Varanasi (DESA, 2018).*

5. **Page 2, line 37: there is a spare "(June to September)" here**
   Removed as suggested.

6. **Page 6, line 138: sorry if I've missed this point elsewhere, but it would be useful to state somewhere in the Methods how the land cover is described / defined in WRF-Chem as this will have an important impact on the biogenic emissions being generated by MEGAN**

   The referee is indeed right, we didn't specify in the manuscript how land cover is described. Our apologies. We used land use and soil categories interpolated from MODIS IGPB 21-category data at 30 arc-seconds resolution (∼ 1 km) (Friedl et al., 2010). We added this information in the Methods:

   - Section 2.1: *For the description of terrain data for the domain (land use and soil categories) we use MODIS IGPB 21 category data at 30 arc-seconds resolution (∼ 1 km) (Friedl et al., 2010).*

7. **7 Page 7, line 162: check this reference?**

   There was an error in the latex bibliography file, we corrected that and now the citation is correct as (Ministry of Environment Government of Pakistan, 2009).

8. **Page 8, line 203-204: could you include a map (in the Appendix?) to show the location of the stations used in the evaluation (those from OpenAQ and the literature values), this would help the reader to understand how they are distributed across the IGP region and the extent to which they can be used to constrain the model's performance in each region**

We included the following map in the appendix showing both ground based stations for PM and gases and for OA from literature. We also added the number ID column in table A2 and A4 corresponding to the number reported in the map.

[Figure]

Figure B1: Location of ground based observation for PM and gases (purple) and OA (orange). ID Number for each station correspond to ID number in table A2 and A4 respectively. The inset map magnifies the area of Delhi NCT.

9. **Page 8, line 206: correct "components"**
   Amended as suggested.

10. **10 Page 8, line 220: somewhere in this section it's worth being clear about the fact that the model has been run for 2017/2018 and many of the observations that are available do not necessarily cover the same time period**

    We added a clarification on this issue in responding to comment 16 of Reviewer 1:

    - Section 2.1: *Data for Pakistan are not available for our modelling study period (2017/2018) so we instead use data from 2019 for the monsoon and postmonsoon seasons and data from 2020 for the winter and pre-monsoon seasons, which represents an additional source of error.*

11. **Page 9, line 224: you say that the poor model skill may be attributed to difficulties in retrieving AOD during the monsoon season, but the model also performs poorly at simulating PM2.5 concentrations during the monsoon season (according to Table A3). Could you edit this section to reflect the fact that, whilst there may be difficulties in retrieving AOD during the monsoon, the model may also not simulate AOD accurately during this period?**

    We added this additional point:

    - Section 2.3: *Poor model skill during the monsoon season may reflect difficulties in retrieving AOD during extensive seasonal cloud coverage and simulating atmospheric aerosols, as highlighted earlier in this section.*

12. **Page 11, line 270: correct "out" to "our"**
    Amended as suggested.

13. **Page 12, line 308: "The sensitivity of PM2.5 is highest for biogenic emissions" this isn't necessarily clear from Figure 4 due to the different color scales used to show the different sources, can you add some quantification to this?**

    This was also a request from reviewer 1. We re-plotted figures 4,7, A4, A5 with a common colourscale. We also added a quantification in the results as suggested:

    - Section 3.2: *We find that $PM_{2.5}$ is sensitive to biogenic emissions over localized regions across the IGP, where $PM_{2.5}$ can be more sensitive to changes in biogenic emissions than to anthropogenic emissions ($\sim$200-500 $\mu g\ m^{-3}$) and <200 $\mu g\ m^{-3}$, respectively).*

14. **Page 12, line 311: add here that this is the simulated / modelled composition (since you do also have observations in the study)**

    We added this clarification as follow:

    - Section 3.2: *Figure 5 shows the model composition of PM2.5 across the IGP.*

**Response to Editor Comments**

**Through personal communication, the Editor asked to specify in the manuscript which years we examined the AOD retrieved from MODIS Terra and Aqua, since this information is not found in the text.**

The editor is indeed right, we didn't specify in the manuscript which years we examined the AOD retrive from MODIS. Our apologies. We compared simulated AOD with satellite retrival of 2017/2018 corresponding to our simulation period. We added this information in the text:

- Appendix B: *We compare our model prediction against satellite AOD retrievals for 2017/2018 at 550 nm with a 10 km horizontal resolution obtained from both Terra (MOD04_L2) and Aqua (MYD04_L2) MODIS instruments.*

- Caption Table A5: *Seasonal comparison of modeled total AOD column and satellite AOD observation for the simulated period 2017/2018.*